# SimMMDG: A Simple and Effective Framework for Multi-modal Domain Generalization

**Hao Dong**[1]  **Ismail Nejjar**[2]  **Han Sun**[2]  **Eleni Chatzi**[1]  **Olga Fink**[2]

[1]ETH Zürich  [2]EPFL

{hao.dong, chatzi}@ibk.baug.ethz.ch, {ismail.nejjar, han.sun, olga.fink}@epfl.ch

## Abstract

In real-world scenarios, achieving domain generalization (DG) presents significant challenges as models are required to generalize to unknown target distributions. Generalizing to unseen multi-modal distributions poses even greater difficulties due to the distinct properties exhibited by different modalities. To overcome the challenges of achieving domain generalization in multi-modal scenarios, we propose *SimMMDG*, a simple yet effective multi-modal DG framework. We argue that mapping features from different modalities into the same embedding space impedes model generalization. To address this, we propose splitting the features within each modality into modality-specific and modality-shared components. We employ supervised contrastive learning on the modality-shared features to ensure they possess joint properties and impose distance constraints on modality-specific features to promote diversity. In addition, we introduce a cross-modal translation module to regularize the learned features, which can also be used for missing-modality generalization. We demonstrate that our framework is theoretically well-supported and achieves strong performance in multi-modal DG on the EPIC-Kitchens dataset and the novel Human-Animal-Cartoon (HAC) dataset introduced in this paper. Our source code and HAC dataset are available at https://github.com/donghao51/SimMMDG.

## 1   Introduction

Domain generalization (DG) has received significant attention in the research community [66]. In real-world scenarios such as autonomous driving [13, 26], robotics [17], action recognition [16], and fault diagnosis [22], it is crucial that models trained on limited source domains perform well across novel target domains. To tackle distribution shift problems, numerous DG algorithms have been proposed, including domain-invariant feature learning [51], feature disentanglement [56], data augmentation [72], and meta-learning [41]. However, most of these algorithms are designed for unimodal data, such as images [40] and time series data [49]. More recently, the emergence of large-scale multi-modal datasets [16, 7] has highlighted the need to address multi-modal DG across multiple modalities, such as audio-video [36, 73], image-language [58, 33], and LiDAR-camera [44, 18]. However, to date, only RNA-Net [57] has focused on the multi-modal DG problem, with a relative norm alignment loss proposed to balance the feature norms of audio and video.

Multi-modal DG is closely related to multi-modal representation learning [47]. Traditional multi-modal contrastive learning frameworks [58, 43] aim to project the features of different modalities into a common embedding space. However, this approach may not be optimal as different modalities consist of both shared information that is consistent across modalities and unique information that is specific to each modality. Consequently, attempting to align all modalities together can be challenging and impractical. For instance, consider descriptions of the same object that use entirely different modalities, such as video, audio, and optical flow, as illustrated in Fig. 1 (a). All modalities share some

37th Conference on Neural Information Processing Systems (NeurIPS 2023).

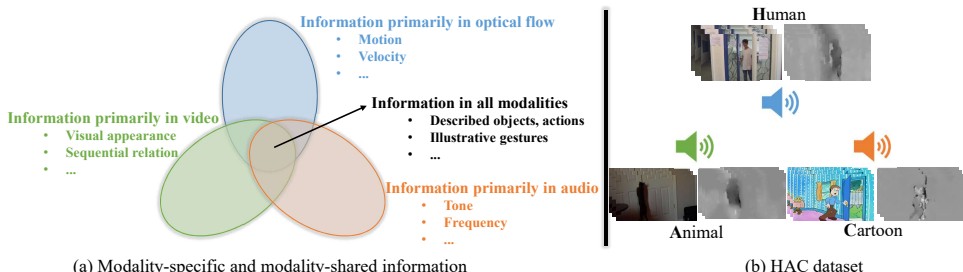

(a) Modality-specific and modality-shared information      (b) HAC dataset

Figure 1: (a). Different modalities possess shared information, while simultaneously containing unique information exclusive to each modality. Inspired by this, we propose to split the feature of each modality into modality-specific and modality-shared parts in our framework. (b) Our new multi-modal DG dataset consists of three domains and three modalities. For each domain, the actor of different actions (opening door in this case) could be either a human, animal, or cartoon figure.

information since they describe the same object, but each modality further provides modality-specific information. For example, videos typically convey visual appearance information and sequential relations between frames, while audio provides tone information that is closely linked to sentiment and frequency information for different sounds. Optical flow, on the other hand, offers more intuitive information on motions and associated velocities. Therefore, simply mapping features from different modalities into the same embedding space would preserve only modality-shared information while overlooking modality-specific details. This could result in the loss of modality-specific information and a decline in downstream task performance.

Effective multi-modal DG frameworks that can address the challenges outlined above are urgently needed. To this end, we propose *SimMMDG* - a **Sim**ple yet effective **M**ulti-**M**odal **DG** framework that can be applied to two or more modalities. We first propose to split the feature embedding of each modality into modality-specific and modality-shared parts, ensuring that complementary information from different modalities is retained. We apply supervised contrastive learning on modality-shared features and incorporate distance constraints for modality-specific features to ensure that the learned representations are informative. Moreover, we introduce a cross-modal translation module to further regularize the learned features and facilitate missing-modality generalization. Some theoretical insights from both the multi-modal representation learning perspective and the domain generalization perspective are provided to demonstrate that our approach is well-motivated in theory. Finally, we release a novel **H**uman-**A**nimal-**C**artoon (HAC) dataset, to promote research in multi-modal domain generalization, as shown in Fig. 1 (b). Our contributions can be summarized as:

- We propose a universal framework for multi-modal domain generalization that demonstrates both simplicity and effectiveness and outperforms existing methods on several challenging datasets. We also demonstrate its efficacy in general multi-modal classification setups.
- We address the missing-modality generalization problem and propose a cross-modal translation module as a solution, which is robust even in scenarios where multiple modalities are missing.
- We provide theoretical insights in support of the efficacy of our proposed approach.
- We introduce a new multi-modal dataset that includes three modalities and large domain shifts, which can serve as a challenging benchmark for future research on multi-modal DG.

## 2   Related Work

**Domain Generalization** refers to the process of training a model on multiple source domains in order to generalize to unseen target domains. This task is more challenging than domain adaptation (DA) [67] as target domain data cannot be accessed during training. Prior research has identified three main categories of domain generalization methods [66], including data manipulation, representation learning, and learning strategies. Data manipulation techniques aim to improve generalization performance by increasing the diversity of training data. For example, Tobin *et al.* [61] use simulated environments for additional data generation, while Zhou *et al.* [75] train a domain transformation network using adversarial training to synthesize data from previously unseen domains. Mixup [72] generates new instances by performing linear interpolations between data and labels. Representation

learning methods seek to learn domain-invariant representations through the use of domain-adversarial neural networks [25, 42], explicit feature distribution alignment [63], and instance normalization [54]. Other approaches utilize learning strategies to improve the generalization performance. Examples of such approaches include meta-learning [41], gradient operation [31], and self-supervised learning [8].

**Multi-modal Representation Learning** aims to learn robust representations through two or more modalities, which can then be applied to various downstream tasks. While unified models [68, 4] tokenize various input modalities into sequences [3] and employ a single Transformer [65] for joint learning, methods such as CLIP [58] and ALIGN [33] use separate encoders for each modality and leverage contrastive loss to align features. However, these methods align features from different modalities into the same embedding space and may only preserve modality-shared information. In contrast, we propose splitting the features of each modality into modality-specific and modality-shared components to ensure that complementary information from different modalities is preserved. A recent work [34] also proposes to split the features into different components in multi-modal representation learning and provide an information-theoretical analysis. However, they only deal with two modalities and without considering missing-modality cases. They also don't use the label information within a batch in contrastive learning.

**Multi-modal DA and DG.** Several approaches exist for multi-modal DA. For instance, Munro and Damen [52] propose a self-supervised alignment approach along with adversarial alignment for multi-modal DA, while Kim *et al.* [38] leverage cross-modal contrastive learning to align cross-modal and cross-domain representations. Zhang *et al.* [73] propose an audio-adaptive encoder and an audio-infused recognizer to address domain shifts. Notably, RNA-Net [57] is the only method known to address multi-modal DG problem, by introducing a relative norm alignment loss to balance audio and video feature norms.

# 3 Methodology

## 3.1 Problem Setting: Multi-modal Domain Generalization

We commence by presenting the definition of the multi-modal domain generalization problem based on the definition of unimodal DG, as described in [66]. Let $\mathbf{X}$ represent a nonempty input space, and $\mathcal{Y}$ denote an output space. A domain, denoted as $\mathcal{D}$, consists of data sampled from a distribution, $\mathcal{D} = \{(\mathbf{x}_j, y_j)\}_{j=1}^n \sim P_{XY}$, where $P_{XY}$ denotes the joint distribution of input samples and output labels. $X$ and $Y$ represent the corresponding random variables.

**Definition 1** (Multi-modal domain generalization). *In multi-modal domain generalization, we are given $D$ training domains $\mathcal{D}_{train} = \{\mathcal{D}^i \mid i = 1, \cdots, D\}$, where $\mathcal{D}^i = \{(\mathbf{x}_j^i, y_j^i)\}_{j=1}^{n_i}$ denotes the i-th domain with $n_i$ data instances. Each data instance $\mathbf{x}_j^i = \{(\mathbf{x}_j^i)_k \mid k = 1, \cdots, M\} \in \mathbf{X}$ is comprised of $M$ different modalities and $y_j^i \in \mathcal{Y} \subset \mathbb{R}$ denotes the label. The joint distributions between each pair of domains are different: $P_{XY}^i \neq P_{XY}^j, 1 \leq i \neq j \leq D$. The goal of multi-modal domain generalization is to learn a robust and generalizable predictive function $f : \mathbf{X} \to \mathcal{Y}$ from $D$ training domains and $M$ data modalities to achieve a minimum prediction error on an* unseen *test domain $\mathcal{D}_{test}$ (i.e., $\mathcal{D}_{test}$ cannot be accessed during training and $P_{XY}^{test} \neq P_{XY}^i$ for $i \in \{1, \cdots, D\}$):*

$$\min_f \mathbb{E}_{(\mathbf{x}, y) \in \mathcal{D}_{test}}[\ell(f(\mathbf{x}), y)], \tag{1}$$

*where $\mathbb{E}$ is the expectation and $\ell(\cdot, \cdot)$ is the loss function.*

## 3.2 SimMMDG

We present the *SimMMDG* framework for addressing the multi-modal DG problem, as depicted in Fig. 2. Our approach involves splitting the feature embedding of each modality into modality-specific and modality-shared components. We aim to map modality-shared embeddings of data samples with the same label, whether they belong to the same or different modalities, to be as close as possible (and vice versa for samples with different labels). For modality-specific features within each modality, our objective is to maximize the distance from their modality-shared features to capture unique and complementary information. Additionally, we incorporate a cross-modal translation module to regularize learned features and address the missing-modality generalization problem, which is essential for real-world testing scenarios where one or more modalities may be absent due to unpredictable factors.

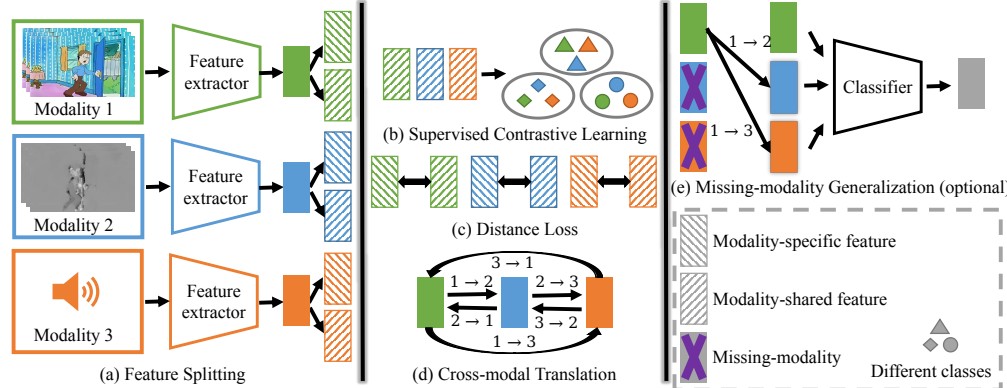

Figure 2: Overview of *SimMMDG*. We split the features of each modality into modality-specific and modality-shared parts. For the modality-shared part, we use supervised contrastive learning to map the features with the same label to be as close as possible. For modality-specific features, we use a distance loss to encourage them to be far from modality-shared features, promoting diversity within each modality. Additionally, we introduce a cross-modal translation module that regularizes features and enhances generalization across missing modalities.

### 3.2.1   Within-modal Feature Splitting

Traditional multi-modal learning frameworks [58, 43] aim to map features from various modalities into a common embedding space using contrastive learning. However, this approach may not be optimal because different modalities often contain a mix of both homogeneous and heterogeneous information, making it challenging and impractical to align them all together. For instance, consider the language and visual modality [2]. Both modalities can describe people, objects, actions, and gestures, which reflect shared information between them. However, images can provide additional information such as texture, depth, and visual appearance that are not available in language data. Similarly, language can provide syntactic structure, vocabulary, and morphology, which are not present in image data. These are specific pieces of information that are unique to each modality.

Thus, the simple mapping of features from different modalities into the same embedding space may result in the loss of modality-specific information and consequently lead to decreased performance in downstream tasks. Guided by this intuition, we propose a novel approach that involves the separation of the feature embedding of each modality into modality-specific and modality-shared components. For example, given a unimodal embedding $\mathbf{E}$, we denote it as $\mathbf{E} = [\mathbf{E}_s; \mathbf{E}_c]$, where $\mathbf{E}_s$ is modality-specific feature and $\mathbf{E}_c$ is modality-shared feature.

**Multi-modal Supervised Contrastive Learning.** We expect modality-shared features $\mathbf{E}_c$ to possess characteristics that are shared among distinct modalities. If data instances from different modalities have the same label, we expect their modality-shared features to be as close as possible in the embedding space. To achieve this objective, we leverage the availability of labels for each data instance and employ supervised contrastive loss [37] for effective guidance. For a set of $N$ randomly sampled label pairs, $\{\boldsymbol{x}_j, y_j\}_{j=1,...,N}$, the corresponding batch used for training consists of $M \times N$ pairs, $\{\tilde{\boldsymbol{x}}_k, \tilde{y}_k\}_{k=1,...,M \times N}$, where $\tilde{\boldsymbol{x}}_{M \times j}, \tilde{\boldsymbol{x}}_{M \times j-1}, ... , \tilde{\boldsymbol{x}}_{M \times j-M+1}$ are data instances from $M$ different modalities in $\boldsymbol{x}_j$ ($j = 1, ..., N$) and $\tilde{y}_{M \times j} = ... = \tilde{y}_{M \times j-M+1} = y_j$.

Let $i \in I \equiv \{1, ..., M \times N\}$ be the index of an arbitrary unimodal sample within a batch. We define $A(i) \equiv I \setminus \{i\}$, $P(i) \equiv \{p \in A(i) : \tilde{y}_p = \tilde{y}_i\}$ as the set of indices of all positive samples in the batch which share the same label as $i$. The cardinality of $P(i)$ is denoted as $|P(i)|$. Then, the multi-modal supervised contrastive learning loss can be written as follows:

$$\mathcal{L}_{con} = \sum_{i \in I} \frac{-1}{|P(i)|} \sum_{p \in P(i)} \log \frac{\exp\left(\boldsymbol{z}_i \cdot \boldsymbol{z}_p / \tau\right)}{\sum_{a \in A(i)} \exp\left(\boldsymbol{z}_i \cdot \boldsymbol{z}_a / \tau\right)}, \tag{2}$$

with $\boldsymbol{z}_k = Proj(g(\tilde{\boldsymbol{x}}_k)) \in \mathcal{R}^{D_P}$, where $g(\cdot)$ is the feature extractor, that maps $\boldsymbol{x}$ to modality-specific and modality-shared features, $\mathbf{E} = [\mathbf{E}_s; \mathbf{E}_c] = g(\boldsymbol{x})$, where $\mathbf{E}_s, \mathbf{E}_c \in R^{D_E}$, and $Proj(\cdot)$ is the

projection network that maps $\mathbf{E}_c$ to a vector $\boldsymbol{z} = Proj(\mathbf{E_c}) \in \mathcal{R}^{D_P}$. The inner product between two projected feature vectors is denoted by $\bullet$, and $\tau \in \mathcal{R}^+$ is a scalar temperature parameter.

**Feature Splitting with Distance.** To ensure that the modality-specific features $\mathbf{E}_s$ carry unique and complementary information, we aim to maximize their dissimilarity from the corresponding modality-shared features $\mathbf{E}_c$. To achieve this goal, we utilize negative $\ell_2$ distance and formulate a distance loss on $\mathbf{E}_s$ and $\mathbf{E}_c$ as:

$$\mathcal{L}_{dis} = \frac{-1}{M} \sum_{i=1}^{M} ||\mathbf{E}_s^i - \mathbf{E}_c^i||_2^2, \tag{3}$$

where $M$ is the number of modalities, $\mathbf{E}_s^i$ and $\mathbf{E}_c^i$ are the modality-specific and modality-shared features of the $i$-th modality.

### 3.2.2 Cross-modal Translation

Simply increasing the distance between $\mathbf{E}_s$ and $\mathbf{E}_c$ may not yield optimal results. The proposed cross-modal translation module aims to ensure the meaningfulness of modality-specific features by exploiting the implicit relationships and approximate translation mappings that exist between the $M$ modalities within the same data instance. This is achieved through a multi-layer perceptron (MLP) [28] that translates the feature embedding $\mathbf{E}$ across modalities. For instance, $\mathbf{E}_t^j = MLP_{\mathbf{E}^i \rightarrow \mathbf{E}^j}(\mathbf{E}^i)$ implies that we translate the embedding $\mathbf{E}^i = [\mathbf{E}_s^i; \mathbf{E}_c^i]$ of the $i$-th modality to the $j$-th modality using $MLP_{\mathbf{E}^i \rightarrow \mathbf{E}^j}$, resulting in the translated embedding $\mathbf{E}_t^j$. More intuitions behind the cross-modal translation module are discussed in the appendix. To ensure that the translated embedding $\mathbf{E}_t^j$ is a meaningful representation of the $j$-th modality, we aim to minimize its $\ell_2$ distance from the real embedding of the $j$-th modality, $\mathbf{E}^j$ and the cross-modal translation loss is defined as:

$$\mathcal{L}_{trans} = \frac{1}{M(M-1)} \sum_{i=1}^{M} \sum_{j \neq i} ||MLP_{\mathbf{E}^i \rightarrow \mathbf{E}^j}(\mathbf{E}^i) - \mathbf{E}^j||_2^2. \tag{4}$$

### 3.3 Final Loss

The final loss is obtained as the weighted sum of the previously defined losses:

$$\mathcal{L} = \mathcal{L}_{cls} + \alpha_{con}\mathcal{L}_{con} + \alpha_{dis}\mathcal{L}_{dis} + \alpha_{trans}\mathcal{L}_{trans}, \tag{5}$$

where $\mathcal{L}_{cls}$ is the cross-entropy loss for classification, and where $\alpha_{\text{con}}$, $\alpha_{\text{dis}}$, and $\alpha_{\text{trans}}$ are hyperparameters that control the relative importance of the contrastive learning, dissimilarity, and cross-modal translation terms, respectively.

### 3.4 Missing-modality Generalization

During the training phase, we have access to data from all modalities. However, in the testing phase, one or more modalities may be absent due to unpredictable reasons, such as sensor malfunction or loss of communication. In such cases, the system's robustness toward potential missing modalities becomes critical. To address the missing-modality scenario, we utilize our cross-modal translation module. In normal circumstances, the output of the network is defined as:

$$y = f(x) = h(g(x)) = h([\mathbf{E}^1, ..., \mathbf{E}^i, ..., \mathbf{E}^M]), \tag{6}$$

where $g(\cdot)$ is the feature extractor and $h(\cdot)$ is the classifier. If the $i$-th modality is missing during testing, one solution is to replace its embedding with zero. The output is then given as:

$$y = h([\mathbf{E}^1, ..., \mathbf{0}, ..., \mathbf{E}^M]). \tag{7}$$

However, simply replacing the embeddings with null entries may have an adverse effect on the network's performance. To address this issue, we propose using our cross-modal translation module to predict and substitute the missing modality's embedding with information from available modalities, resulting in a more robust output for the network. The output of the network is then given as:

$$y = h([\mathbf{E}^1, ..., \mathbf{E}_t^i, ..., \mathbf{E}^M]), \tag{8}$$

where

$$\mathbf{E}_t^i = \frac{1}{M-1} \sum_{j \neq i}^{M} MLP_{\mathbf{E}^j \to \mathbf{E}^i}(\mathbf{E}^j). \tag{9}$$

In cases where multiple modalities are missing during testing, we can use the same strategy to utilize available modalities to predict and substitute the missing ones, similar to Eq. (9). The benefits of our approach are demonstrated in the subsequent experiments.

## 4 Theoretical Insights

### 4.1 Multi-modal Representation Learning Perspective

We first generalize Theorem 3.1 in [34] to the case of $M$ modalities. Let $p$ be the joint distribution of $(x_1, ..., x_M, y)$, where $x_i$ is the $i$-th modality and $y$ is the target. We define the *information gap* as $\Delta_p := \max\{I(x_i; y); i = 1, ..., M\} - \min\{I(x_i; y); i = 1, ..., M\}$, where $I(x_i; y)$ is the mutual information between modality $x_i$ and the target variable $y$. The information gap serves to characterize the effectiveness of the modalities in predicting the target $y$. We use the cross-entropy loss, denoted by $\ell_{\text{CE}}$, and the prediction function, denoted by $f$.

**Theorem 1.** *For $M$ feature extractors $g^i(\cdot)$ ($i = 1, ..., M$), if the multi-modal features $\mathbf{E}^i = g^i(x_i)$ are perfectly aligned in the feature space, i.e., $\mathbf{E}^1 = ... = \mathbf{E}^i = ... = \mathbf{E}^M$, then $\inf_f \mathbb{E}_p[\ell_{CE}(f(\mathbf{E}^1, ..., \mathbf{E}^M), y)] - \inf_{f'} \mathbb{E}_p[\ell_{CE}(f'(x_1, ..., x_M), y)] \geq \Delta_p$.*

The proof of this theorem can be found in the appendix. Theorem 1 indicates that the optimal prediction error, achieved with perfectly aligned features, is at least $\Delta_p$ larger than when using the raw input modalities. In practical scenarios where one modality is less informative for prediction, the information gap $\Delta_p$ tends to be substantial. Consequently, this leads to a notable increase in prediction errors in downstream tasks. Furthermore, achieving perfect modality alignment enforces the aligned features to exclusively contain predictive information present in all input modalities, potentially causing the loss of modality-specific information. By splitting the feature embedding for each modality, our model has access to both modality-shared features with predictive information present in both modalities, and modality-specific features that contain predictive information unique to each individual modality. This enables our model to effectively capture the distinct information provided by each modality, while simultaneously leveraging the shared information across modalities. Consequently, our approach has the potential to enhance generalization capabilities and achieve better performance on downstream tasks.

### 4.2 Domain Generalization Perspective

**Theorem 2.** *Let $\mathcal{X}$ be a space and $\mathcal{H}$ be a class of hypotheses corresponding to this space. Let $\mathbb{Q}$ and the collection $\{\mathbb{P}_i\}_{i=1}^K$ be distributions over $\mathcal{X}$ and let $\{\varphi_i\}_{i=1}^K$ be a collection of non-negative coefficient with $\sum_i \varphi_i = 1$. Let $\mathcal{O}$ be a set of distributions s.t. $\forall \mathbb{S} \in \mathcal{O}$, the following holds*

$$\sum_i \varphi_i d_{\mathcal{H}\Delta\mathcal{H}}(\mathbb{P}_i, \mathbb{S}) \leq \max_{i,j} d_{\mathcal{H}\Delta\mathcal{H}}(\mathbb{P}_i, \mathbb{P}_j). \tag{10}$$

*Then, for any $h \in \mathcal{H}$,*

$$\varepsilon_{\mathbb{Q}}(h) \leq \lambda_\varphi + \sum_i \varphi_i \varepsilon_{\mathbb{P}_i}(h) + \frac{1}{2} \min_{\mathbb{S} \in \mathcal{O}} d_{\mathcal{H}\Delta\mathcal{H}}(\mathbb{S}, \mathbb{Q}) + \frac{1}{2} \max_{i,j} d_{\mathcal{H}\Delta\mathcal{H}}(\mathbb{P}_i, \mathbb{P}_j), \tag{11}$$

*where $\lambda_\varphi$ is the error of an ideal joint hypothesis, $\varepsilon_{\mathbb{P}}(h)$ is the error for a hypothesis $h$ on a distribution $\mathbb{P}$, and $d_{\mathcal{H}\Delta\mathcal{H}}(\mathbb{P}, \mathbb{Q})$ is $\mathcal{H}$-divergence which measures differences in distribution [6].*

The proof of this theorem is provided in [60]. Here, $\mathbb{Q}$ corresponds to the unseen out-of-distribution target domain and $\{\mathbb{P}_i\}_{i=1}^K$ correspond to source domains. $\lambda_\varphi$ is small in reality and often neglected. The term $\sum_i \varphi_i \varepsilon_{\mathbb{P}_i}(h)$ is minimized by cross-entropy loss with class labels as supervision. The term $\frac{1}{2} \max_{i,j} d_{\mathcal{H}\Delta\mathcal{H}}(\mathbb{P}_i, \mathbb{P}_j)$ measures the maximum differences among source domains. This corresponds to multi-modal supervised contrastive learning in our approach, where we map the modality-shared features of different modalities from all source domains to be as close as possible

if they have the same label. The term $\frac{1}{2}\min_{\mathbb{S}\in\mathcal{O}} d_{\mathcal{H}\Delta\mathcal{H}}(\mathbb{S}, \mathbb{Q})$ demonstrates the importance of diverse source distributions [1], so that the unseen target $\mathbb{Q}$ might be "near" to $\mathcal{O}$. Therefore, to enlarge the range of $\mathcal{O}$, we split the feature embedding of each modality into modality-specific and modality-shared parts by maximizing the $\ell_2$ distance to preserve diversity.

## 5 Experiments

### 5.1 Experimental Setting

**Dataset.** We use the EPIC-Kitchens dataset [16] and introduce a novel HAC dataset in this paper, which will be made publicly accessible for further research. We follow the experimental protocol used for the EPIC-Kitchens dataset in [52]. The EPIC-Kitchens dataset includes eight actions ('put', 'take', 'open', 'close', 'wash', 'cut', 'mix', and 'pour') recorded in three different kitchens, forming three separate domains D1, D2, and D3. Our HAC dataset consists of seven actions ('sleeping', 'watching tv', 'eating', 'drinking', 'swimming', 'running', and 'opening door') performed by humans, animals, and cartoon figures, forming three different domains H, A, and C. We collect 3381 video clips from the internet with around 1000 samples for each domain. We provide three modalities in our dataset: video, audio, and pre-computed optical flow. Some examples of the HAC dataset are shown in Fig. 1 (b) and more details are provided in the supplementary material.

**Implementation Details.** In our framework, we perform experiments on three modalities: video, audio, and optical flow. We adopt the MMAction2 [15] toolkit for experiments. To encode the visual information, we use SlowFast network [21] initialized with Kinetics-400 [35] pre-trained weights. For the audio encoder, we use ResNet-18 [29] and initialize the weights from the VGGSound pre-trained checkpoint [12]. Similarly, we use the SlowFast network with slow-only pathway and also Kinetics 400 [35] pre-trained weights for the optical flow encoder. The dimensions of the unimodal embedding **E** for video, audio, and optical flow are 2304, 512, and 2048 correspondingly. For the projection network $Proj(\cdot)$ in supervised contrastive learning, we instantiate it as a multi-layer perceptron with two hidden layers of size 2048 and output vector of size $D_P = 128$. We use a multi-layer perceptron with two hidden layers of size 2048 to instantiate the cross-modal translation $MLP_{\mathbf{E}^i \rightarrow \mathbf{E}^j}$. After obtaining the feature embedding from the encoder, we split the embedding into modality-shared features (the first half of the embedding) and modality-specific features (the remaining half). We use the Adam optimizer [39] with a learning rate of 0.0001 and a batch size of 16. The scalar temperature parameter $\tau$ is set to 0.1. Additionally, we set $\alpha_{con} = 3.0$, $\alpha_{dis} = 0.7$, and $\alpha_{trans} = 0.1$. We also analyze the sensitivity of different $\alpha$ in the appendix. Finally, we train the network for 15 epochs on an RTX 2080 Ti GPU which takes about 20 hours and select the model with the best performance on the validation dataset. We report the Top-1 accuracy for all experiments.

### 5.2 Results

**Multi-modal Multi-source DG.** Tab. 1 and Tab. 2 illustrate the results of *SimMMDG* under the multi-modal multi-source DG setting, where we train on multiple source domains and test on one target domain. We first conduct experiments using video and audio modalities for experiments, as in [57]. We re-implement our framework employing the same I3D [10] and BN-Inception [32] backbone for video and audio as in RNA-Net [57] to ensure fair comparisons. The DeepAll approach implies that we feed all the data from source domains to the network without any domain generalization strategies. Tab. 1 shows that our *SimMMDG* significantly outperforms all the baselines. When we replace the backbone with SlowFast [21] and ResNet-18 [29],

| Method | D2, D3 → D1 | D1, D3 → D2 | D1, D2 → D3 | *Mean* |
|---|---|---|---|---|
| **I3D backbone** | | | | |
| DeepAll | 43.19 | 39.35 | 51.47 | 44.67 |
| IBN-Net [53] | 44.46 | 49.21 | 48.97 | 47.55 |
| Gradient Blending [69] | 41.97 | 48.40 | 51.43 | 47.27 |
| TBN [36] | 42.35 | 47.45 | 49.20 | 46.33 |
| AVSA [50] | 42.78 | 47.38 | 51.79 | 47.32 |
| Co-Attention [14] | 40.87 | 43.57 | 54.88 | 46.44 |
| SENet [30] | 42.82 | 42.81 | 51.07 | 45.56 |
| Non-Local [70] | 45.72 | 43.08 | 49.49 | 46.10 |
| MM-SADA [52] | 39.79 | 52.73 | 51.87 | 48.13 |
| RNA-Net [57] | 45.65 | 51.64 | 55.88 | 51.06 |
| SimMMDG (ours) | **54.25** | **58.67** | **57.28** | **56.73** |
| **SlowFast backbone** | | | | |
| DeepAll | 47.13 | 55.73 | 57.17 | 53.34 |
| MM-SADA [52] | 49.20 | 60.40 | 59.14 | 56.25 |
| RNA-Net [57] | 52.18 | 59.47 | 60.88 | 57.51 |
| SimMMDG (ours) | **57.93** | **65.47** | **66.32** | **63.24** |

Table 1: Multi-modal **multi-source** DG on EPIC-Kitchens dataset using video and audio.

| Method | Modality | | | EPIC-Kitchens dataset | | | | HAC dataset | | | |
|---|---|---|---|---|---|---|---|---|---|---|---|
| | Video | Audio | Flow | D2, D3 → D1 | D1, D3 → D2 | D1, D2 → D3 | Mean | A, C → H | H, C → A | H, A → C | Mean |
| DeepAll | ✓ | ✓ | | 47.13 | 55.73 | 57.17 | 53.34 | 66.55 | 72.85 | 45.77 | 61.72 |
| MM-SADA [52] | ✓ | ✓ | | 49.20 | 60.40 | 59.14 | 56.25 | 65.47 | 72.52 | 44.30 | 60.76 |
| RNA-Net [57] | ✓ | ✓ | | 52.18 | 59.47 | 60.88 | 57.51 | 60.20 | 73.95 | 48.90 | 61.02 |
| SimMMDG (ours) | ✓ | ✓ | | **57.93** | **65.47** | **66.32** | **63.24** | **74.77** | **77.81** | **53.68** | **68.75** |
| DeepAll | ✓ | | ✓ | 55.17 | 62.93 | 60.37 | 59.49 | 76.78 | 70.64 | 49.63 | 65.68 |
| MM-SADA [52] | ✓ | | ✓ | 47.13 | 57.60 | 59.34 | 54.69 | 69.79 | 69.76 | 49.45 | 63.00 |
| RNA-Net [57] | ✓ | | ✓ | 54.71 | 61.87 | 58.21 | 58.26 | 77.14 | 74.94 | 42.00 | 64.69 |
| SimMMDG (ours) | ✓ | | ✓ | **59.31** | **63.33** | **62.73** | **61.79** | **79.31** | **77.04** | **51.29** | **69.21** |
| DeepAll | | ✓ | ✓ | 45.28 | 56.40 | 57.08 | 52.92 | 50.04 | 59.71 | 38.97 | 49.57 |
| MM-SADA [52] | | ✓ | ✓ | 47.36 | 53.47 | 60.27 | 53.70 | 46.58 | 61.81 | 39.15 | 49.18 |
| RNA-Net [57] | | ✓ | ✓ | 45.74 | 57.73 | 56.47 | 53.31 | 52.05 | 64.13 | 40.35 | 52.18 |
| SimMMDG (ours) | | ✓ | ✓ | **56.09** | **67.33** | **61.50** | **61.64** | **59.63** | **64.24** | **44.85** | **56.24** |
| DeepAll | ✓ | ✓ | ✓ | 55.63 | 59.20 | 58.01 | 57.61 | 69.07 | 71.30 | 51.47 | 63.95 |
| MM-SADA [52] | ✓ | ✓ | ✓ | 51.72 | 58.40 | 59.34 | 56.49 | 72.53 | 72.19 | 55.51 | 66.74 |
| RNA-Net [57] | ✓ | ✓ | ✓ | 52.41 | 57.20 | 60.16 | 56.59 | 69.00 | 73.40 | 51.65 | 64.68 |
| SimMMDG (ours) | ✓ | ✓ | ✓ | **63.68** | **70.13** | **67.76** | **67.19** | **77.65** | **79.03** | **56.62** | **71.10** |

Table 2: Multi-modal **multi-source** DG with different modalities on EPIC-Kitchens and HAC datasets.

| Method | | EPIC-Kitchens dataset | | | | | | | HAC dataset | | | | | | |
|---|---|---|---|---|---|---|---|---|---|---|---|---|---|---|---|
| | Source: | D1 | | D2 | | D3 | | | H | | A | | C | | |
| Method | Target: | D2 | D3 | D1 | D3 | D1 | D2 | Mean | A | C | H | C | H | A | Mean |
| DeepAll | | 52.27 | 51.75 | 44.60 | 54.11 | 48.05 | 56.67 | 51.24 | 66.11 | 43.01 | 63.45 | 37.68 | 46.86 | 58.94 | 52.68 |
| MM-SADA [52] | | 50.67 | 50.10 | 51.49 | 56.57 | 42.99 | 54.00 | 50.97 | 65.89 | 37.22 | 57.75 | 40.90 | 49.82 | 62.91 | 52.42 |
| RNA-Net [57] | | 45.60 | 46.30 | 43.68 | 57.39 | 49.66 | 55.87 | 49.75 | 65.67 | 42.92 | 61.72 | 38.69 | 47.66 | 61.59 | 53.04 |
| SimMMDG (ours) | | **54.00** | **52.26** | **51.49** | **60.88** | **49.88** | **60.53** | **54.84** | **66.67** | **44.21** | **68.42** | **46.05** | **54.43** | **72.73** | **58.75** |

Table 3: Multi-modal **single-source** DG on EPIC-Kitchens and HAC datasets using video and audio.

the results further improve by a large margin (with an average improvement of up to $5.73\%$) compared to the baselines. In the following experiments, we adopt SlowFast and ResNet-18 as our default backbones. To verify the generalization of our framework to different modalities, we conduct experiments by combining any two modalities, as well as all three modalities, and present the results in Tab. 2. Our *SimMMDG* outperforms all the baselines by a significant margin in all cases, with improvements of up to $9.58\%$. Notably, when we combine all three modalities, the performance further improves and surpasses that of any two modalities. In contrast, the baseline methods cannot achieve better results with more modalities, indicating that they fail to fully leverage the complementary information between modalities. Finally, we validate the performance of our framework on the HAC dataset, and the results are consistent with those obtained on the EPIC-Kitchens dataset, as demonstrated in Tab. 2. Our *SimMMDG* outperforms all the baselines by a significant margin in all cases, with improvements of up to $7.73\%$.

**Multi-modal Single-source DG.** The domain labels are not required in our *SimMMDG* framework. This feature makes our method readily applicable to single-source DG without modifications. In this setup, we train the model on a single source domain and test it on multiple target domains. Tab. 3 presents the results of *SimMMDG* under the multi-modal single-source DG setting using video and audio modalities. Despite being trained only on data from a single domain, our model demonstrates robust generalization to unseen domains, with an average improvement of up to $5.71\%$. In comparison, other baseline methods perform even worse than the DeepAll baseline, highlighting their limitations in the single-source DG setting. We present additional results obtained by exploring different combinations of modalities in the appendix.

**Missing-modality DG.** In real-world deployment scenarios, we cannot always guarantee that all modalities will be available. Hence, the network should be capable of handling missing-modality cases. One simple approach is to set the embedding of the missing modality to zero. We compare this with our proposed cross-modal translation replacement in Sec. 3.4. We retrain the cross-modal translation $MLP_{\mathbf{E}^i \rightarrow \mathbf{E}^j}$ for 10 epochs using $\mathcal{L}_{trans}$ in Eq. (4) and keep other parameters fixed. Tab. 4 reports on the comparison results on the EPIC-Kitchens dataset, where our solution yields significant benefits (up to $10.47\%$ performance improvement) compared to zero-filling. By replacing the missing modality features with the translation ones, our approach also outperforms the unimodal model in most cases. In contrast, zero-filling hurts the network performance in some cases and is even worse than the unimodal model. The last six rows in Tab. 4 demonstrate that our approach is *robust*

| Method | Video | Audio | Flow | D2, D3 → D1 | D1, D3 → D2 | D1, D2 → D3 | *Mean* |
|---|---|---|---|---|---|---|---|
| DeepAll (Video-only) | ✓ | | | 51.03 | 59.87 | 56.57 | 55.82 |
| SimMMDG (zero-filling) | ✓ | ✗ | | 55.40 | 64.00 | 58.32 | 59.24 |
| SimMMDG (translation) | ✓ | ✗ | | **57.24** | **65.07** | **59.65** | **60.65** |
| SimMMDG (zero-filling) | ✓ | | ✗ | 53.10 | 60.93 | 61.09 | 58.37 |
| SimMMDG (translation) | ✓ | | ✗ | **55.17** | **61.33** | **61.70** | **59.40** |
| DeepAll (Audio-only) | | ✓ | | 32.87 | 42.27 | 45.17 | 40.10 |
| SimMMDG (zero-filling) | ✗ | ✓ | | 29.20 | 37.33 | 45.69 | 37.41 |
| SimMMDG (translation) | ✗ | ✓ | | **37.70** | **46.40** | **52.05** | **45.38** |
| SimMMDG (zero-filling) | | ✓ | ✗ | 28.51 | 34.27 | 42.81 | 35.20 |
| SimMMDG (translation) | | ✓ | ✗ | **39.77** | **46.00** | **51.23** | **45.67** |
| DeepAll (Flow-only) | | | ✓ | 54.25 | 61.33 | 55.95 | 57.18 |
| SimMMDG (zero-filling) | ✗ | | ✓ | 56.32 | 60.67 | 56.57 | 57.85 |
| SimMMDG (translation) | ✗ | | ✓ | **56.32** | **62.13** | **56.98** | **58.48** |
| SimMMDG (zero-filling) | | ✗ | ✓ | 52.41 | 62.13 | 51.33 | 55.29 |
| SimMMDG (translation) | | ✗ | ✓ | **55.40** | **63.73** | **56.57** | **58.57** |
| SimMMDG (zero-filling) | ✗ | ✓ | ✓ | 53.10 | 62.40 | 64.48 | 59.99 |
| SimMMDG (translation) | ✗ | ✓ | ✓ | **55.86** | **68.27** | **64.58** | **62.90** |
| SimMMDG (zero-filling) | ✓ | ✗ | ✓ | 62.76 | 66.80 | 57.49 | 62.35 |
| SimMMDG (translation) | ✓ | ✗ | ✓ | **63.22** | **67.20** | **59.24** | **63.22** |
| SimMMDG (zero-filling) | ✓ | ✓ | ✗ | 60.92 | 68.66 | 60.99 | 63.52 |
| SimMMDG (translation) | ✓ | ✓ | ✗ | **62.53** | 68.13 | **61.60** | **64.09** |
| SimMMDG (zero-filling) | ✓ | ✗ | ✗ | 59.08 | 62.27 | 52.98 | 58.11 |
| SimMMDG (translation) | ✓ | ✗ | ✗ | **59.31** | **64.00** | **56.57** | **59.96** |
| SimMMDG (zero-filling) | ✗ | ✓ | ✗ | 33.79 | 37.47 | 44.56 | 38.61 |
| SimMMDG (translation) | ✗ | ✓ | ✗ | **38.39** | **44.53** | **47.74** | **43.55** |
| SimMMDG (zero-filling) | ✗ | ✗ | ✓ | 55.17 | 57.07 | 48.46 | 53.57 |
| SimMMDG (translation) | ✗ | ✗ | ✓ | **56.32** | **64.27** | **56.57** | **59.05** |

Table 4: Multi-modal multi-source DG with **missing modalities** on EPIC-Kitchens dataset. ✗ means the modality is available during training, but is missing in test time.

*even when two modalities out of three are missing*. We present more results on HAC dataset in the appendix.

### 5.3 Ablation Studies

**Ablation on each proposed module.** We conducted extensive ablation studies to investigate the role of each module of *SimMMDG* on EPIC-Kitchens dataset, as shown in Tab. 5. Incorporating the supervised contrastive learning loss alone resulted in noticeable improvements. However, the mean accuracy decreased when we integrated feature splitting without imposing any constraints. The results were further enhanced when we added the distance loss to promote diversity, and even more so when we incorporated the cross-modal translation module. Although using only supervised contrastive learning and cross-modal translation yielded satisfactory results, their performance is on average $4.15\%$ lower than the complete approach with feature splitting and distance loss. These findings highlight the significance of segregating the feature embedding of each modality into modality-specific and modality-shared components. Although *SimMMDG* without supervised contrastive learning is already better than most baselines, it still has a performance gap compared with the whole framework, which means the cross-modal translation is helpful but cannot replace contrastive learning. The effect of contrastive learning is similar to explicit feature alignment. It aligns the modality-shared features of different modalities from different source domains with the same label to be as close as possible in the embedding space, while pushing away features with different labels, to make the embedding space more distinctive.

**Comparison against unimodal DG.** Tab. 6 presents the results in comparison to unimodal DG algorithms that exclusively rely on either video, audio, or optical flow inputs. We choose RSC [31], Mixup [72], and Fishr [59] as our baselines. By leveraging information from multiple modalities, our multi-modal DG framework delivers significant improvements (up to $7.74\%$) in terms of performance as compared to unimodal DG methods.

**Combine *SimMMDG* with other training strategies.** *SimMMDG* can be seamlessly combined with other training strategies due to its generality and simplicity. We first combine *SimMMDG* with Gradient Blending [69], a strategy to improve multi-modal learning. The results are further improved compared to the original *SimMMDG* as shown in Tab. 7. We also combine *SimMMDG* with a Domain-adversarial Neural Network (DANN) [24] to align the features of source domains using

| CL | FS | DL | CT | D2, D3 → D1 | D1, D3 → D2 | D1, D2 → D3 | *Mean* |
|----|----|----|----|-------------|-------------|-------------|--------|
|    |    |    |    | 55.86 | 56.27 | 54.21 | 55.45 |
| ✓  |    |    |    | 52.18 | 61.60 | 62.22 | 58.67 |
| ✓  |    |    | ✓  | 51.49 | 62.53 | 63.24 | 59.09 |
| ✓  | ✓  |    |    | 51.72 | 62.67 | 58.93 | 57.77 |
| ✓  | ✓  | ✓  |    | 55.17 | 64.00 | 64.37 | 61.18 |
|    | ✓  | ✓  | ✓  | 54.25 | 63.47 | 43.04 | 60.25 |
| ✓  | ✓  | ✓  | ✓  | **57.93** | **65.47** | **66.32** | **63.24** |

Table 5: Ablations of each proposed module on EPIC-Kitchens dataset. CL: supervised contrastive learning, FS: feature splitting, DL: distance loss, CT: cross-modal translation.

| Method | D2, D3 → D1 | D1, D3 → D2 | D1, D2 → D3 | *Mean* |
|--------|-------------|-------------|-------------|--------|
| RSC (V) | 50.11 | 62.53 | 58.73 | 57.12 |
| RSC (A) | 36.55 | 43.73 | 48.15 | 42.81 |
| RSC (F) | 55.17 | 63.33 | 59.65 | 59.38 |
| Mixup (V) | 49.20 | 59.73 | 59.96 | 56.30 |
| Mixup (A) | 35.17 | 40.80 | 45.07 | 40.35 |
| Mixup (F) | 56.32 | 65.60 | 54.62 | 58.85 |
| Fishr (V) | 53.79 | 63.47 | 61.09 | 59.45 |
| Fishr (A) | 37.47 | 44.80 | 47.43 | 43.23 |
| Fishr (F) | 54.25 | 63.87 | 59.14 | 59.09 |
| SimMMDG (V+A+F) | **63.68** | **70.13** | **67.76** | **67.19** |

Table 6: Comparison with single-modal DG methods on EPIC-Kitchens dataset. V: video, A: audio, F: optical flow.

| Method | D2, D3 → D1 | D1, D3 → D2 | D1, D2 → D3 | *Mean* |
|--------|-------------|-------------|-------------|--------|
| SimMMDG | 57.93 | 65.47 | 66.32 | 63.24 |
| +Gradient Blending | 59.31 | 68.40 | 66.63 | **64.78** |
| +DANN | 60.69 | 66.69 | 64.58 | **64.07** |

Table 7: Combine SimMMDG with other training strategies on EPIC-Kitchens dataset.

| Method | EPIC-Kitchens | Method | MUSTARD | UR-FUNNY |
|--------|---------------|--------|---------|----------|
| DeepAll | 64.9 | Late Fusion | 61.6 | 63.6 |
| TBN | 73.2 | LMF | 65.2 | 63.9 |
| Gradient Blending | 74.0 | MULT | 60.9 | 63.2 |
| SimMMDG | **76.4** | SimMMDG | **72.5** | **65.6** |

Table 8: Evaluation under multi-modal classification setup.

domain labels and observe a similar improvement as with Gradient Blending. This indicates that our *SimMMDG* can be easily combined with other training strategies to get even better results.

***SimMMDG* as a general framework for multi-modal classification.** Here, we evaluate our framework in a more general multi-modal classification setup without DG. We first evaluate our framework on the EPIC-Kitchens dataset without incorporating the DG setup. To do so, we aggregate data from three domains, we partition the training, validation, and testing data to ensure that no apparent domain shifts exit between the training and testing data. We compare our method with DeepAll, TBN [36], and Gradient Blending [69]. The results, as presented in Tab. 8 demonstrate that *SimMMDG* also exhibits significant advantages in the general multi-modal classification setup.

We further evaluate our framework on two multi-modal datasets: MUSTARD [11] and UR-FUNNY [27], both available in MultiBench [45]. These datasets pertain to human sentiment analysis and encompass language, video, and audio modalities. We conduct our experiments using the Multi-Bench codebase and implement *SimMMDG* within that environment. MultiBench treats human sentiment analysis as a regression task, whereas our framework is tailored for classification. To align the codebase with our classification task, we made the necessary modifications. For all baselines, we apply the same backbone model and solely change the fusion paradigms. In our comparisons, we evaluate our method against Late Fusion, Low-rank Tensor Fusion (LMF) [48], and Multimodal Transformer Fusion (MULT) [62]. The results reveal that our *SimMMDG* exhibits clear advantages, outperforming the baselines with an average improvement of 7.3% and 1.7%. This suggests that our *SimMMDG* serves as a versatile framework for multi-modal classification tasks and is compatible with various combinations of modalities, such as video+audio+flow and language+video+audio.

## 6   Conclusion

In this paper, we propose the *SimMMDG* framework for multi-modal DG. Our approach involves splitting the features of each modality into modality-specific and modality-shared parts and enforcing constraints on each part using supervised contrastive learning, distance loss, and cross-modal translation. The cross-modal translation module can also be applied in the case of missing-modality generalization. Our experiments on different datasets demonstrate the effectiveness of *SimMMDG*. Furthermore, we introduce a new challenging multi-modal dataset that can serve as a benchmark and guide future research in multi-modal DG problems.

**Limitations.** Currently, the number of cross-modal translation MLP in our framework is $\mathcal{O}(|M|^2)$ and will be complex with the increase of modalities. In future work, the encoder-decoder network proposed in [19] can be used to reduce the complexity to $\mathcal{O}(|M|)$.

## Acknowledgments

The authors acknowledge the support of "In-service diagnostics of the catenary/pantograph and wheelset axle systems through intelligent algorithms" (SENTINEL) project, supported by the ETH Mobility Initiative.

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

# A   Proof of Theorem 1

*Proof of Theorem 1.* Consider the joint mutual information $I(\mathbf{E}^1, ..., \mathbf{E}^M; y)$. By the chain rule, we have the following decompositions:

$$
\begin{aligned}
I(\mathbf{E}^1, ..., \mathbf{E}^M; y) &= I(\mathbf{E}^1; y) + I(\mathbf{E}^2, ..., \mathbf{E}^M; y \mid \mathbf{E}^1) \\
&= ... \\
&= I(\mathbf{E}^M; y) + I(\mathbf{E}^1, ..., \mathbf{E}^{M-1}; y \mid \mathbf{E}^M).
\end{aligned}
$$

However, since $\mathbf{E}^i$ are perfectly aligned, $I(\mathbf{E}^2, ..., \mathbf{E}^M; y \mid \mathbf{E}^1) = ... = I(\mathbf{E}^1, ..., \mathbf{E}^{M-1}; y \mid \mathbf{E}^M) = 0$, which means $I(\mathbf{E}^1, ..., \mathbf{E}^M; y) = I(\mathbf{E}^1; y) = ... = I(\mathbf{E}^M; y)$. On the other hand, by the data processing inequality [5], we know that

$$
I(\mathbf{E}^1; y) \leq I(x_1; y), ..., I(\mathbf{E}^M; y) \leq I(x_M; y).
$$

Hence, the following chain of inequalities holds:

$$
\begin{aligned}
I(\mathbf{E}^1, ..., \mathbf{E}^M; y) &= \min\{I(\mathbf{E}^1; y), ..., I(\mathbf{E}^M; y)\} \\
&\leq \min\{I(x_1; y), ..., I(x_M; y)\} \\
&\leq \max\{I(x_1; y), ..., I(x_M; y)\} \\
&\leq I(x_1, ..., x_M; y),
\end{aligned}
$$

where the last inequality follows from the fact that the joint mutual information $I(x_1, ..., x_M; y)$, is at least as large as any one of $I(x_i; y)$. We use $H(y \mid x_1, ..., x_M)$ to denote the conditional entropy of $y$ given $x_1, ..., x_M$ as input. According to [74, 20] we have the following variational form: $H(y \mid x_1, ..., x_M) = \inf_f \mathbb{E}_p[\ell_{\mathrm{CE}}(f(x_1, ..., x_M), y)]$, where the infimum is over all the prediction functions that take $x_1, ..., x_M$ as input to predict the target $y$ and the expectation is taken over the joint distribution $p$ of $(x_1, ..., x_M, y)$. Therefore, due to the variational form of the conditional entropy, we have

$$
\begin{aligned}
&\inf_f \mathbb{E}_p[\ell_{\mathrm{CE}}(f(\mathbf{E}^1, ..., \mathbf{E}^M), y)] - \inf_{f'} \mathbb{E}_p[\ell_{\mathrm{CE}}(f'(x_1, ..., x_M), y)] \\
&= H(y \mid \mathbf{E}^1, ..., \mathbf{E}^M) - H(y \mid x_1, ..., x_M) \\
&= H(y) - H(y \mid x_1, ..., x_M) - (H(y) - H(y \mid \mathbf{E}^1, ..., \mathbf{E}^M)) \\
&= I(x_1, ..., x_M; y) - I(\mathbf{E}^1, ..., \mathbf{E}^M; y) \\
&\geq \max\{I(x_1; y), ..., I(x_M; y)\} - \min\{I(x_1; y), ..., I(x_M; y)\} \\
&= \max\{I(x_i; y); i = 1, ..., M\} - \min\{I(x_i; y); i = 1, ..., M\} \\
&= \Delta_p. \qquad \square
\end{aligned}
$$

# B   More Details on HAC Dataset

Our Human-Animal-Cartoon (HAC) dataset consists of seven actions ('sleeping', 'watching tv', 'eating', 'drinking', 'swimming', 'running', and 'opening door') performed by humans, animals, and cartoon figures, forming three different domains. We collect 3381 video clips from the internet with around 1000 for each domain and provide three modalities in our dataset: video, audio, and pre-computed optical flow. The dense optical flow is extracted at 24 frames per second using the TV-L1 algorithm [71].

Our dataset was collected by 5 volunteers. For the human domain, we collect the data by selecting actions from an existing Kinetics-600 dataset [9]. We select approximately the same number of video clips for each action to ensure class balance. For the animal domain, we use the available 200 video clips in [73] and extend it to 906. We collect data from YouTube by searching keywords like 'animal sleeping', 'animal eating', 'animal running', etc. To increase the diversity of the dataset, we also specify the animal type in the keywords, such as 'cat sleeping', 'dog eating', 'horse running', etc. Each participant was asked to collect certain actions, such as Participant A for 'sleeping' and 'watching tv', Participant B for 'eating' and 'drinking', etc. For the cartoon domain, we collect all data from scratch and we collect data from popular cartoons like 'SpongeBob SquarePants', 'The Simpsons', 'Garfield and Friends', etc. Each participant was asked to collect from one or two cartoon

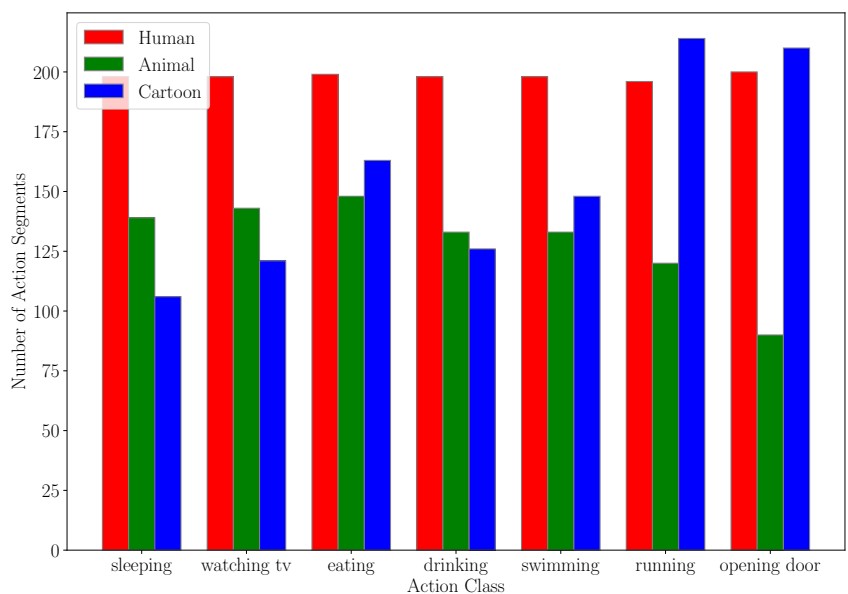

Figure 3: The number of each action segments for our HAC dataset.

| Domain | Human | Animal | Cartoon |
|---|---|---|---|
| Training Action Segments | 1111 | 730 | 870 |
| Validation Action Segments | 276 | 176 | 218 |
| Testing Action Segments | 1387 | 906 | 1088 |

Table 9: Number of action segments per domain.

series to avoid duplication. For each action, we annotated the start and end times in the video and then cut out video clips. The length of each video clip varies from 1 s to 10 s. Finally, we gathered the data from all volunteers and a separate person manually discarded unqualified data like duplicate videos, videos without audio data, noisy/wrong classes, etc.

Fig. 3 and Tab. 9 illustrate the statistcs of our HAC dataset. During the training phase, we train our model on one or two domains by utilizing the training action segments specific to those domains. The model selection is based on the performance of the validation action segments. Subsequently, we evaluate the model on the remaining domain, utilizing all the segments within that domain. Therefore, the total number of testing action segments is the sum of the training and validation segments. It is observed that each class within the HAC dataset exhibits a good balance across all domains. In contrast, EPIC-Kitchens dataset suffers from severe class imbalance. This implies that our proposed framework is capable of delivering high-performance results across both balanced and imbalanced datasets. Fig. 4 presents additional examples from our HAC dataset, highlighting the substantial domain shifts present in our dataset, particularly in the cartoon domain. The main purpose of this dataset is to be used for multi-modal domain generalization research. Of course, our dataset can also be used for other applications like multi-modal learning and multi-modal domain adaptation.

## C   Additional Implementation Details

We follow RNA-Net [57] to select the baseline methods. MM-SADA [52] is a domain adaption method and we only use the self-supervision loss without adversarial alignment as we have no access on target domain data during training. From the results in Table 1 in main paper, we know RNA-Net [57] and MM-SADA [52] are the two best baseline methods. Therefore, we select them together with DeepAll as baselines in the following experiments.

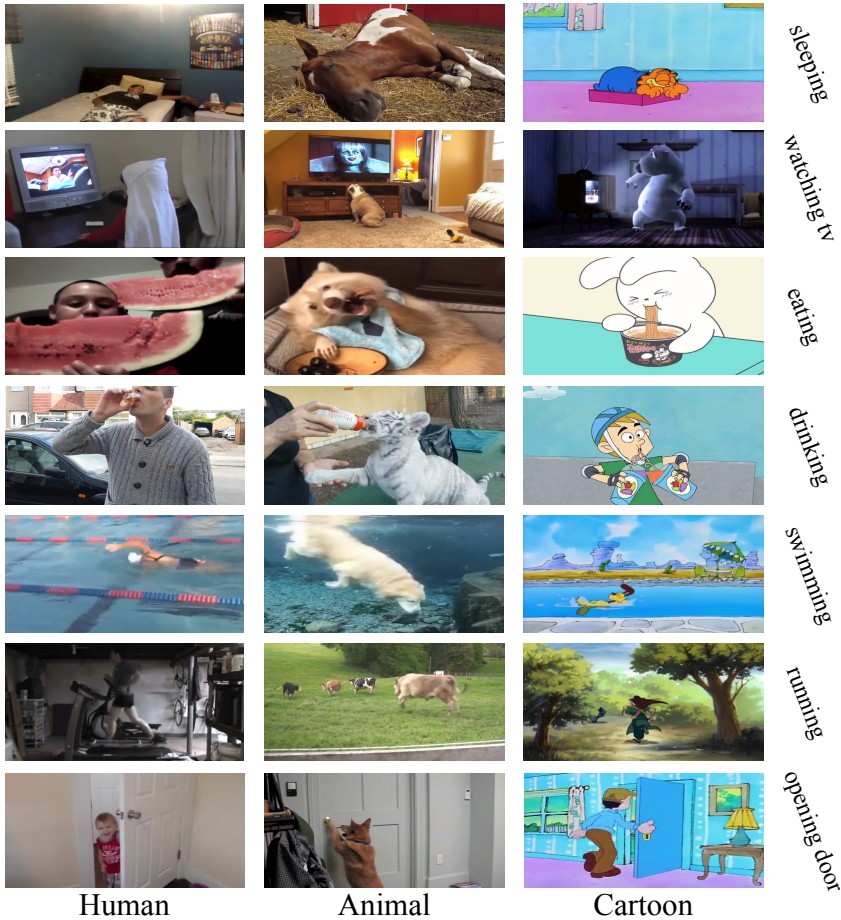

Figure 4: More examples on our HAC dataset.

| Distance Type | D2, D3 → D1 | D1, D3 → D2 | D1, D2 → D3 | *Mean* |
|---|---|---|---|---|
| Cosine | 53.56 | 60.80 | 59.65 | 58.00 |
| $\ell_1$ norm | 55.86 | 62.27 | 58.73 | 58.95 |
| $\ell_2$ norm | **57.93** | **65.47** | **66.32** | **63.24** |

Table 10: Ablation of different distance types in distance loss on EPIC-Kitchens dataset.

# D  Further Ablations

**Ablations of different distance types.** Tab. 10 shows the ablations of different distance types in distance loss $\mathcal{L}_{dis}$. We explore the effects of using $\ell_2$ norm, $\ell_1$ norm, and Cosine similarity. Our experimental results indicate that all three distance types outperform the baselines, with $\ell_2$ norm achieving the highest performance.

**Parameter Sensitivity.** We investigate the sensitivity of our method to the hyperparameters in the loss function. We perform this analysis by varying one parameter while fixing the others, and present our findings in Fig. 5. The results demonstrate that our method consistently outperforms the DeepAll baseline for all parameter settings, indicating that our approach is less sensitive to hyperparameter choices. However, we can also observe $\alpha_{dis}$ is a little sensitive and may need attention for tuning in real applications.

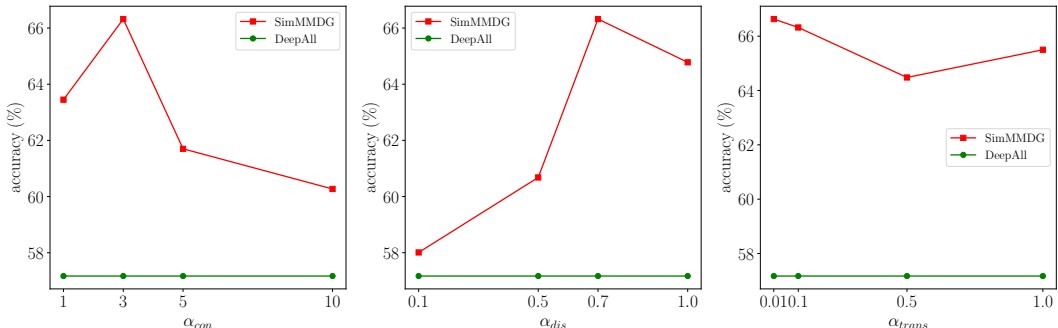

Figure 5: Parameter sensitivity (D1, D2 → D3 in EPIC-Kitchens for multi-modal multi-source DG).

| | Modality | | | Source: D1 | | Source: D2 | | Source: D3 | | |
|---|---|---|---|---|---|---|---|---|---|---|
| Method | Video | Audio | Flow | D1→ D2 | D1→ D3 | D2 → D1 | D2 → D3 | D3→ D1 | D3→ D2 | *Mean* |
| DeepAll | ✓ | | ✓ | 53.07 | 47.74 | 49.66 | 56.98 | **52.18** | 55.73 | 52.56 |
| MM-SADA [52] | ✓ | | ✓ | 49.20 | 48.87 | 51.26 | 59.03 | 44.60 | 56.40 | 51.56 |
| RNA-Net [57] | ✓ | | ✓ | 53.20 | 49.79 | 52.18 | 58.11 | 47.13 | 55.47 | 52.65 |
| SimMMDG (ours) | ✓ | | ✓ | **56.80** | **54.11** | **53.10** | **59.86** | 50.57 | **64.27** | **56.45** |
| DeepAll | | ✓ | ✓ | 41.60 | 47.43 | 41.61 | 51.75 | 45.98 | 56.13 | 47.42 |
| MM-SADA [52] | | ✓ | ✓ | 42.53 | 47.64 | 42.76 | 50.92 | 42.07 | 51.87 | 46.30 |
| RNA-Net [57] | | ✓ | ✓ | 46.40 | 49.38 | 42.76 | 54.62 | 43.45 | 52.13 | 48.12 |
| SimMMDG (ours) | | ✓ | ✓ | **50.00** | **49.49** | **45.98** | **56.26** | **51.95** | **63.87** | **52.93** |
| DeepAll | ✓ | ✓ | ✓ | 51.60 | 53.39 | 44.60 | **60.16** | 45.74 | 53.47 | 51.49 |
| MM-SADA [52] | ✓ | ✓ | ✓ | 49.43 | 48.53 | 38.16 | 48.46 | 44.14 | 51.07 | 46.63 |
| RNA-Net [57] | ✓ | ✓ | ✓ | 51.20 | 53.90 | 46.67 | 58.73 | 49.43 | 57.47 | 52.90 |
| SimMMDG (ours) | ✓ | ✓ | ✓ | **55.33** | **54.83** | **50.80** | 59.45 | **53.33** | **64.40** | **56.36** |

Table 11: Multi-modal **single-source** DG with different modalities on EPIC-Kitchens dataset.

| | Modality | | | Source: H | | Source: A | | Source: C | | |
|---|---|---|---|---|---|---|---|---|---|---|
| Method | Video | Audio | Flow | H→ A | H→ C | A → H | A → C | C→ H | C→ A | *Mean* |
| DeepAll | ✓ | | ✓ | 59.82 | 41.08 | 73.25 | 35.02 | 60.71 | 64.46 | 55.72 |
| MM-SADA [52] | ✓ | | ✓ | **66.23** | **46.23** | 69.65 | 42.46 | 59.77 | 57.62 | 56.99 |
| RNA-Net [57] | ✓ | | ✓ | 58.61 | 37.87 | 73.90 | 48.16 | 60.35 | 59.38 | 56.38 |
| SimMMDG (ours) | ✓ | | ✓ | 63.13 | 44.58 | **74.62** | **51.38** | **70.15** | **61.59** | **60.91** |
| DeepAll | | ✓ | ✓ | 59.93 | 35.29 | 50.90 | 35.20 | 32.73 | 39.85 | 42.32 |
| MM-SADA [52] | | ✓ | ✓ | 55.08 | 32.90 | 52.85 | 34.10 | 29.70 | 44.48 | 41.52 |
| RNA-Net [57] | | ✓ | ✓ | 56.51 | 30.24 | 53.50 | 33.46 | 36.91 | 40.40 | 41.84 |
| SimMMDG (ours) | | ✓ | ✓ | **62.25** | **36.40** | **56.60** | **38.79** | **38.72** | **48.12** | **46.81** |
| DeepAll | ✓ | ✓ | ✓ | 62.58 | 42.56 | 67.34 | 46.05 | 53.86 | 59.49 | 55.31 |
| MM-SADA [52] | ✓ | ✓ | ✓ | 62.25 | 44.67 | 60.78 | 43.11 | 41.82 | 44.81 | 49.57 |
| RNA-Net [57] | ✓ | ✓ | ✓ | 66.89 | 45.40 | 66.62 | 46.88 | 53.64 | 61.37 | 56.80 |
| SimMMDG (ours) | ✓ | ✓ | ✓ | **68.76** | **47.98** | **70.94** | **49.26** | **65.18** | **62.69** | **60.80** |

Table 12: Multi-modal **single-source** DG with different modalities on HAC dataset.

# E    Other Experimental Results

**Multi-modal Single-source DG.** Tab. 11 and Tab. 12 show more results under multi-modal single-source DG setting using the combination of different modalities. Our model demonstrates robust generalization to unseen domains compared to the baseline methods in most cases with a large margin up to 4.81%.

**Missing-modality DG.** Tab. 13 shows more results under the missing-modality DG setting on HAC dataset. Our solution yields significant benefits compared to zero-filling in most cases, similar to findings in the EPIC-Kitchens dataset. Our approach is also robust even when two modalities out of three are missing. We also validate the effectiveness of our cross-modal translation module on DeepAll, MM-SADA [52], and RNA-Net [57], as shown in Tab. 14. Compared to zero-filling,

| Method | Video | Audio | Flow | A, C → H | H, C → A | H, A → C | *Mean* |
|---|---|---|---|---|---|---|---|
| DeepAll (Video-only) | ✓ | | | 75.20 | 74.83 | 38.88 | 62.97 |
| SimMMDG (zero-filling) | ✓ | ✗ | | 73.97 | 75.94 | 51.10 | 67.00 |
| SimMMDG (translation) | ✓ | ✗ | | **77.51** | **77.15** | **53.03** | **69.23** |
| SimMMDG (zero-filling) | ✓ | | ✗ | 79.60 | 76.93 | 50.09 | 68.87 |
| SimMMDG (translation) | ✓ | | ✗ | **79.67** | **77.15** | **51.29** | **69.37** |
| DeepAll (Audio-only) | | ✓ | | 29.99 | 37.75 | 23.35 | 30.36 |
| SimMMDG (zero-filling) | ✗ | ✓ | | 25.16 | 26.05 | 15.99 | 22.40 |
| SimMMDG (translation) | ✗ | ✓ | | **28.26** | **39.74** | **22.15** | **30.05** |
| SimMMDG (zero-filling) | | ✓ | ✗ | 31.36 | 38.63 | 26.65 | 32.21 |
| SimMMDG (translation) | | ✓ | ✗ | **33.81** | **40.95** | **27.02** | **33.93** |
| DeepAll (Flow-only) | | | ✓ | 54.87 | 56.62 | 40.25 | 50.58 |
| SimMMDG (zero-filling) | ✗ | | ✓ | 54.79 | 51.43 | 40.44 | 48.89 |
| SimMMDG (translation) | ✗ | | ✓ | **59.12** | **60.38** | **41.18** | **53.56** |
| SimMMDG (zero-filling) | | ✗ | ✓ | 57.75 | 60.38 | **44.12** | 54.08 |
| SimMMDG (translation) | | ✗ | ✓ | **58.54** | **61.26** | 42.65 | **54.15** |
| SimMMDG (zero-filling) | ✗ | ✓ | ✓ | 47.87 | 51.21 | **41.45** | 46.84 |
| SimMMDG (translation) | ✗ | ✓ | ✓ | **52.05** | **61.70** | 41.08 | **51.61** |
| SimMMDG (zero-filling) | ✓ | ✗ | ✓ | **80.32** | **79.36** | **55.88** | **71.85** |
| SimMMDG (translation) | ✓ | ✗ | ✓ | 79.24 | 78.26 | 53.49 | 70.33 |
| SimMMDG (zero-filling) | ✓ | ✓ | ✗ | 76.64 | 79.14 | 53.31 | 69.70 |
| SimMMDG (translation) | ✓ | ✓ | ✗ | **77.58** | **79.91** | **55.88** | **71.12** |
| SimMMDG (zero-filling) | ✓ | ✗ | ✗ | **80.39** | **79.03** | 52.67 | 70.70 |
| SimMMDG (translation) | ✓ | ✗ | ✗ | 79.60 | 78.48 | **54.04** | **70.71** |
| SimMMDG (zero-filling) | ✗ | ✓ | ✗ | 22.71 | 32.45 | 23.62 | 26.26 |
| SimMMDG (translation) | ✗ | ✓ | ✗ | **30.28** | **41.83** | **29.60** | **33.90** |
| SimMMDG (zero-filling) | ✗ | ✗ | ✓ | 56.02 | 57.40 | 38.97 | 50.80 |
| SimMMDG (translation) | ✗ | ✗ | ✓ | **57.25** | **61.04** | **39.15** | **52.48** |

Table 13: Multi-modal multi-source DG with **missing modalities** on HAC dataset. ✗ means the modality is available during training, but is missing in test time.

| Method | Video | Audio | D2, D3 → D1 | D1, D3 → D2 | D1, D2 → D3 | *Mean* |
|---|---|---|---|---|---|---|
| DeepAll (Video-only) | ✓ | | 51.03 | 59.87 | 56.57 | 55.82 |
| DeepAll (zero-filling) | ✓ | ✗ | 45.98 | 53.73 | 52.87 | 50.86 |
| DeepAll (translation) | ✓ | ✗ | **47.82** | **54.27** | **54.83** | **52.31** |
| MM-SADA [52] (zero-filling) | ✓ | ✗ | 48.97 | 58.00 | 58.52 | 55.16 |
| MM-SADA [52] (translation) | ✓ | ✗ | **50.80** | **58.27** | **58.93** | **56.00** |
| RNA-Net [57] (zero-filling) | ✓ | ✗ | 52.41 | 54.80 | 53.08 | 53.43 |
| RNA-Net [57] (translation) | ✓ | ✗ | **54.25** | **58.13** | **55.85** | **56.08** |
| SimMMDG (zero-filling) | ✓ | ✗ | 55.40 | 64.00 | 58.32 | 59.24 |
| SimMMDG (translation) | ✓ | ✗ | **57.24** | **65.07** | **59.65** | **60.65** |
| DeepAll (Audio-only) | | ✓ | 32.87 | 42.27 | 45.17 | 40.10 |
| DeepAll (zero-filling) | ✗ | ✓ | 37.70 | 37.47 | 46.71 | 40.63 |
| DeepAll (translation) | ✗ | ✓ | **38.39** | **42.13** | **50.10** | **43.54** |
| MM-SADA [52] (zero-filling) | ✗ | ✓ | 36.09 | 40.13 | 46.51 | 40.91 |
| MM-SADA [52] (translation) | ✗ | ✓ | **37.01** | **41.33** | **47.95** | **42.10** |
| RNA-Net [57] (zero-filling) | ✗ | ✓ | 35.86 | 40.27 | 46.71 | 40.95 |
| RNA-Net [57] (translation) | ✗ | ✓ | **39.31** | **42.53** | **48.97** | **43.60** |
| SimMMDG (zero-filling) | ✗ | ✓ | 29.20 | 37.33 | 45.69 | 37.41 |
| SimMMDG (translation) | ✗ | ✓ | **37.70** | **46.40** | **52.05** | **45.38** |

Table 14: Multi-modal multi-source DG with **missing modalities** on EPIC-Kitchens dataset for different baselines. ✗ means the modality is available during training, but is missing in test time.

replacing the features of missing modalities with the translated ones from other available modalities achieves better performance.

**Statistical Significance Tests.** We run each experiment three times using different seeds for multi-modal multi-source DG on the EPIC-Kitchens dataset and then calculate the mean and standard deviation to show the statistical significance of our methods. As shown in Tab. 15, our framework is statistically stable and surpasses the baselines significantly.

| Method | D2, D3 → D1 | D1, D3 → D2 | D1, D2 → D3 | *Mean* |
|---|---|---|---|---|
| DeepAll | $48.81 \pm 1.77$ | $55.78 \pm 1.14$ | $57.45 \pm 1.19$ | $54.01 \pm 0.58$ |
| MM-SADA [52] | $48.89 \pm 0.43$ | $59.15 \pm 1.10$ | $59.31 \pm 0.81$ | $55.79 \pm 0.69$ |
| RNA-Net [57] | $51.26 \pm 3.07$ | $58.54 \pm 0.82$ | $59.14 \pm 1.24$ | $56.31 \pm 1.22$ |
| SimMMDG (ours) | $\mathbf{57.70} \pm 0.50$ | $\mathbf{67.33} \pm 0.99$ | $\mathbf{64.41} \pm 1.36$ | $\mathbf{63.15} \pm 0.88$ |

Table 15: **Statistical significance tests** for multi-modal multi-source DG on EPIC-Kitchens dataset using video and audio.

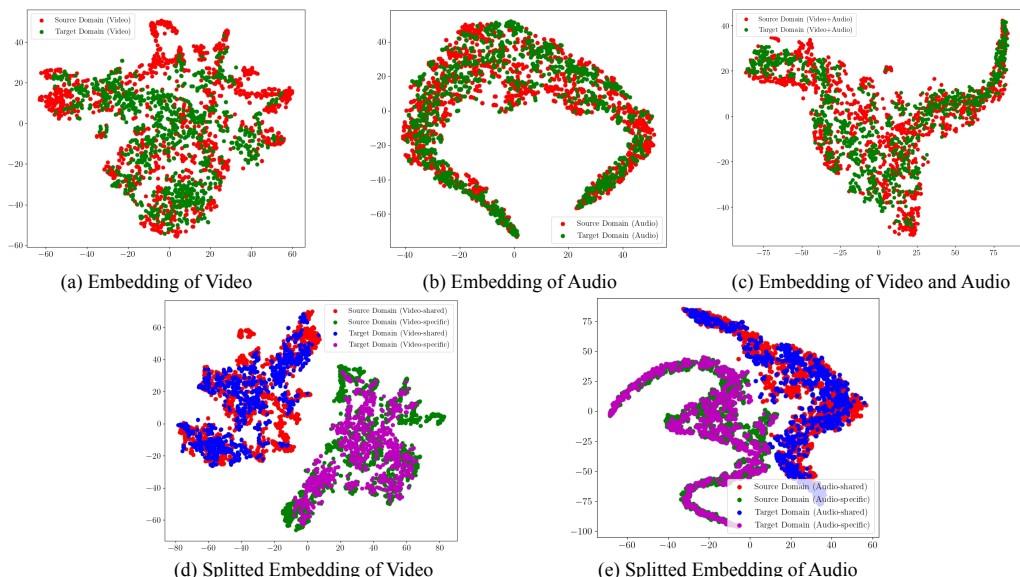

(a) Embedding of Video     (b) Embedding of Audio     (c) Embedding of Video and Audio

(d) Splitted Embedding of Video     (e) Splitted Embedding of Audio

Figure 6: Visualization of the learned embeddings using t-SNE (D1, D2 → D3 in EPIC-Kitchens for multi-modal multi-source DG).

## F    Visualization

We show more visualizations of the learned embeddings using t-SNE [64] in Fig. 6. We can observe that the embeddings of video, audio, and the concatenation of video and audio are all well aligned for the source and target domains. Besides, the modality-specific and modality-shared embeddings are also well-separated and aligned across domains.

## G    More Intuitions behind Cross-modal Translation Module

In our *SimMMDG* framework, we introduce a cross-modal translation module to further regularize the learned features and facilitate missing-modality generalization. Here we give more intuitions behind this module.

**Cross-modal translation won't undermine the unique features of different modalities.** We want to learn an MLP projection to translate the embedding $\mathbf{E}^i$ of the $i$-th modality to the embedding $\mathbf{E}^j$ of the $j$-th modality. We add a translation loss to make the translated embedding $\mathbf{E}^j_t$ to be close to $\mathbf{E}^j$, without any explicit alignment or constraints on $\mathbf{E}^i$ and $\mathbf{E}^j$. We apply the cross-modal translation on the integrated feature of each modality $\mathbf{E}^i = [\mathbf{E}^i_s; \mathbf{E}^i_c]$, which is the concatenation of modality-specific feature $\mathbf{E}_s$ and modality-shared feature $\mathbf{E}_c$. We still enforce a distance loss on $\mathbf{E}_s$ and $\mathbf{E}_c$ at the same time. Therefore, the modality-specific and modality-shared features are still forced to be separated during the training progress. The embedding visualizations shown in Fig. 6 (d) and (e) also indicate that for both video and audio, their modality-specific and modality-shared features are well disentangled and are not influenced by the cross-modal translation module.

| | R@1 | R@5 | R@10 |
|---|---|---|---|
| Modality-specific Features | 11.80 | 45.37 | 49.78 |
| Modality-shared Features | **76.58** | **90.13** | **92.83** |

Table 16: Video to Audio Retrieval.

| | R@1 | R@5 | R@10 |
|---|---|---|---|
| Modality-specific Features | 10.24 | 33.01 | 47.10 |
| Modality-shared Features | **67.59** | **89.60** | **93.99** |

Table 17: Audio to Video Retrieval.

**Cross-modal translation is a type of modalities interaction,** where different modality elements interact to give rise to new information when integrated together for task inference [46]. Several works [23, 55] have already demonstrated that modality interaction can help improve the performance of multi-modal tasks. For example, the proposed approach in [23] learns a coordinated similarity space between image and text to improve image classification. The approach proposed in [55] translates language into video and audio for language sentiment analysis.

**Cross-modal translation can be thought of as a means to leverage the information from one modality to infer as much information as possible for the target modality.** Just like when we hear a dog barking, we will fill in the picture of the puppy in our mind, and when we see a foreign language, we will automatically translate it into our native language. Although there is information loss and we can't recover all the details during this translation progress, we can still infer some useful information for the target modality. As shown in our ablation study in Tab. 5, adding the cross-modal translation module indeed improves multi-modal DG performance.

More importantly, our **cross-modal translation module can be used for improving missing-modality generalization,** by filling in the features of missing modality with the features inferred/translated from the available modality. The benefit of adopting cross-modal translation is demonstrated in Tab. 4. Our approach is robust even when two modalities out of three are missing. This indicates that the inferred information from the cross-modal translation module is valuable and useful for downstream tasks.

## H  Further Discussions on Modality-specific and Modality-shared Features

Based on our assumptions in the paper, modality-shared features reflect shared information between all modalities, like all modalities that describe the same people, objects, actions, or gestures. Modality-specific features are specific pieces of information that are unique to each modality, like texture, depth, and visual appearance in images, syntactic structure, vocabulary, and morphology in language. Our goal is to align the modality-shared features of different modalities from different source domains with the same label to be as close as possible in the embedding space, while pushing away features with different labels, to make the embedding space more distinctive. At the same time, we want the modality-specific features to be as far as possible from the modality-shared features, such that they carry different types of information.

We analyzed the meaningfulness and the ability to share features of modality-shared features by cross-modal retrieval task. The ability of cross-modal retrieval is highly related to the shareable of features, as only features that are meaningful, shareable, and highly connective can give a good recall rate. We use modality-shared features of videos to retrieve from the modality-shared features of audios and calculate the recall, and vice versa for audios. We report the R@1, R@5, and R@10 values for Video to Audio Retrieval and Audio to Video Retrieval. For example, R@5 for Video to Audio Retrieval means we retrieve 5 candidates from audio, if at least one of them has the same label as the query video, we consider the retrieval success and then we calculate the average success rate for all query videos. We also do the same thing on modality-specific features: using modality-specific features of videos to retrieve from the modality-specific features of audios, and vice versa for audios. As shown in Tab. 16 and Tab. 17, we have a very high recall rate when we use modality-shared features, while the recall rate is biased towards random when we use modality-specific features. This further indicates that the modality-shared features are truly shareable and meaningful and there are very strong relations and connections across modalities. The modality-specific features are instead with more information that is private to each single modality.

To further verify the meaningfulness of modality-shared features, we also train a classifier only on the concatenation of modality-shared features of video and audio, and discard the modality-specific features. As shown in Tab. 18, without modality-specific features, the performances drop slightly

| Method | D2, D3 → D1 | D1, D3 → D2 | D1, D2 → D3 | *Mean* |
|---|---|---|---|---|
| SimMMDG (modality-shared features only) | 54.71 | 64.67 | 62.83 | 60.74 |
| SimMMDG | 57.93 | 65.47 | 66.32 | 63.24 |

Table 18: Multi-modal multi-source DG on EPIC-Kitchens dataset using only modality-shared features.

compared to the whole framework, but are still competitive. This indicates that modality-shared features truly have some meaningful information that can be used for prediction tasks.

