# OpenReview forum: "SimMMDG: A Simple and Effective Framework for Multi-modal Domain Generalization"
_NeurIPS.cc/2023/Conference — NeurIPS 2023 poster_

### Official Review · Reviewer_8EtQ · 2023-07-03

**Soundness:** 3 good
**Presentation:** 3 good
**Contribution:** 3 good
**Rating:** 5
**Confidence:** 5

**Summary:**

This work focuses on addressing a special case of multimodal learning, i.e., multi-modal domain generalization, where one or more modalities will be absent during test phase.

This work propose a approach to solve multimodal DG from a disentanglement perspective.  Modality-specific features contain modality-specific information, while Modality-shared feature contains information that is shared across domain such as class labels.

Supervised contrastive learning is employed to learn better modality-shared features while a distance-based regularize is used to enlarge the discrepancy between modality-specific and modality-shared features. In addition, a cross-modal translation module is proposed to enhance the modality-shared features.

This work also release a dataset named Human-Animal-Cartoon (HAC).

**Strengths:**

This work has following strengths:

- This paper released a dataset, HAC, which is helpful for multimodal domain generalization community.

- This paper is easy to follow.

- Multi-modal DG is interesting and has research potential since real-world task is complicated that often involves multiple modalities.

**Weaknesses:**

This work has following weaknesses:

- Is the name of the setting appropriate? Typically, domain generalization refer to a setting that testing domain is unseen during training. However, the setting in this work is that the testing domain/modality exists in training data but some training modalities might be missing.

- Lack of proof that the network truly learn modality-specific and modality-shared features.

- It's not clear that why Cross-modal Translation can enhance modality-specific features, since the MLP use both features for translation. How to guarantee the two types features are well disentangled.

- The technique contribution of this work is somehow not strong. Supervised contrastive loss is a mature technique. What's the essence to use supervised contrastive loss here? Can CE loss obtain similar performance? Translation idea is also appeared in [1].

[1] Ge, Yunhao, et al. "Zero-shot synthesis with group-supervised learning." arXiv preprint arXiv:2009.06586 (2020).

**Questions:**



**Limitations:**

---

> ### Author Rebuttal · Authors · 2023-08-08
>
> Thanks for your insightful reviews, and we appreciate your valuable suggestions! We address your concerns and questions as follows:
> >**Q1**: However, the setting in this work is that the testing domain/modality exists in training data but some training modalities might be missing.
>
> **A1**: In our setting, the testing domain data is completely unseen during training and we only train our model on source domains. This is exactly the same setup as traditional DG, the only difference is that we take multiple modalities as input. In this setup, we train the model on source domains with at least two modalities as input data and test on target domains that have the same modalities but have large distribution shifts compared to source domains.
>
> Benefiting from the simplicity of our framework and the proposed cross-modal translation module, our method can also be used in multi-modal single-source DG and missing-modality DG directly without any modification.
> ___
> >**Q2**: Lack of proof that the network truly learn modality-specific and modality-shared features. How to guarantee the two types features are well disentangled.
>
> **A2**: The feature disentanglement is ensured by the proposed distance loss, where we maximize the dissimilarity between modality-shared and modality-specific features. According to the embedding visualization shown in Figure 4 (d) and (e) in the Supplementary Material, for both video and audio modalities, their modality-specific and modality-shared features are well disentangled. One of the goals of our proposed framework is to maximize the dissimilarity between modality-specific and modality-shared features, which means the network truly learns such properties.
> ___
> >**Q3**: It's not clear that why Cross-modal Translation can enhance modality-specific features.
>
> **A3**:
> 1. **Cross-modal translation is a type of modalities interaction**, where different modality elements interact to give rise to new information when integrated together for task inference [1]. Several works [2,3] have already demonstrated that modalities interaction can help improve the performance of multi-modal tasks.
>
> 2. **Cross-modal translation can be thought of as means to leveraging the information from one modality to infer as much information as possible for the target modality.** Just like when we hear a dog barking, we will fill in the picture of the puppy in our mind, and when we see a foreign language, we will automatically translate it into our native language. Although there is information loss and we can't recover all the details during this translation progress, we can still infer some useful information for the target modality. As shown in our ablation study (Table 5 in the main paper), adding the cross-modal translation module indeed improves multi-modal DG performance.
>
> 3. More importantly, our **cross-modal translation module can be used for improving missing-modality generalization,** by filling in the features of missing modality with the features inferred/translated from the available modality. The benefit of adopting cross-modal translation is demonstrated in Table 4 of the main paper. Our approach is robust even when two modalities out of three are missing. This indicates that the inferred information from the cross-modal translation module is valuable and useful for downstream tasks.
>
> [1] Paul Pu Liang, et al. Foundations and recent trends in multimodal machine learning: Principles, challenges, and open questions, arXiv preprint, 2022.
>
> [2] Andrea Frome, et al. Devise: A deep visual-semantic embedding model. In NeurIPS, 2013.
>
> [3] Hai Pham, et al. Found in translation: Learning robust joint representations by cyclic translations between modalities. In AAAI, 2019.
> ___
>
> >**Q4**: The technique contribution of this work is somehow not strong.
>
> **A4**: Traditional multi-modal learning frameworks use contrastive loss (CL) on the integrated features of different modalities, while we only use CL on modality-shared features and add distance loss on modality-specific features to learn complementary information from different modalities. The effect of CL is similar to explicit feature alignment. It aligns the modality-shared features of different modalities from different source domains with the same label to be as close as possible in the embedding space while pushing away features with different labels, to make the embedding space more distinctive.
>
> We evaluated additionally the contributions of the supervised contrastive loss (CL) and the results are reported below. Although SimMMDG without CL is already better than most baselines, it still has a performance gap compared with our proposed entire framework (which includes CL), which indicates the importance of CL.
>
> |Method|D2,D3→D1|D1,D3→D2|D1,D2→D3|mean|
> |---------|----------|----------|----------|------|
> |SimMMDG w/o CL|54.25|63.47|63.04|60.25|
> |SimMMDG|**57.93**|**65.47**|**66.32**|**63.24**|
>
> The approach proposed in [1] **swaps** disentangled features in the latent space across examples, while we **translate** features from one modality to another. This is a key difference with respect to existing schemes and a main contribution of this work. Moreover, [1] disentangles features at the level of single data samples, e.g. identity, pose, and background. However, we disentangle features at the level of the different modalities, e.g. modality-shared and modality-specific features.
>
> We also address the missing-modality generalization problem with our proposed cross-modal translation module, which is robust even in scenarios where multiple modalities are missing. We also provide theoretical insights in support of the efficacy of our proposed approach (section 4 in the main paper).
>
> We show further findings in the global response that: **SimMMDG is a general framework for multi-modal classification tasks, not limited to domain generalization setup, and can be seamlessly combined with other training strategies to achieve better performances.**

---

> > ### Comment · Reviewer_8EtQ · 2023-08-11
> >
> > Further question regarding Q2.
> > Figure 4 (d) and (e) show that the modality-shared features are aligned and the modality-specific features are aligned. 1. How to make sure that the modality-shared feature is modality-shared,  and the modality-specific feature is modality-specific?

---

> > > ### Author Response · Authors · 2023-08-12
> > >
> > > Thanks for proposing your further questions.
> > >
> > > Based on our assumptions in the paper, modality-shared features reflect shared information between all modalities, like all modalities that describe the same people, objects, actions, or gestures. Modality-specific features are specific pieces of information that are unique to each modality, like texture, depth, and visual appearance in images, syntactic structure, vocabulary, and morphology in language. Our goal is to align the modality-shared features of different modalities from different source domains with the same label to be as close as possible in the embedding space, while pushing away features with different labels, to make the embedding space more distinctive. At the same time, we want the modality-specific features to be as far as possible from the modality-shared features, such that they carry different types of information.
> > >
> > > From the methodology view, we make sure of this by using supervised contrastive loss on modality-shared features and adding distance loss between modality-specific and modality-shared features.
> > >
> > > From the experimental results view, we further calculate two correlation metrics to validate our assumption. Let's assume $\mathbf{E}^{v}_s$ to be the modality-specific feature of video and $\mathbf{E}^{v}_c$ to be the modality-shared feature of video. We define similarly $\mathbf{E}^{a}_s$ and $\mathbf{E}^{a}_c$ for audio. We first calculate the Pearson correlation coefficient between modality-specific and modality-shared features. Pearson correlation coefficient is the most common way of measuring a linear correlation between two sets of data. It is a number between –1 and 1 that measures the strength and direction of the relationship between two variables. From our results, the Pearson correlation coefficient between $\mathbf{E}^{v}_c$ and $\mathbf{E}^{a}_c$ is 0.70, which means the modality-shared features of video and audio have a strong positive correlation. The Pearson correlation coefficient between $\mathbf{E}^{v}_s$ and $\mathbf{E}^{a}_s$ is -0.44, which means the modality-specific features of video and audio have a weak negative correlation. This is consistent with our assumption that modality-specific features reflect shared information and should have a high correlation, while modality-specific features reflect unique information in each modality and should have a weak correlation.
> > >
> > > We also calculate the Cosine similarity between modality-specific and modality-shared features. Cosine similarity is a measure of similarity between two non-zero vectors defined in an inner product space, which is also related to correlation. Cosine similarity is also a number between –1 and 1 that measures the strength and direction of the relationship between two variables. From our results, the Cosine similarity between modality-shared features $\mathbf{E}^{v}_c$ and $\mathbf{E}^{a}_c$ is 0.82 and the Cosine similarity between modality-specific features $\mathbf{E}^{v}_s$ and $\mathbf{E}^{a}_s$ is -0.10. This also indicates that the modality-shared features of video and audio have a strong positive correlation and the modality-specific features of video and audio have a weak negative correlation.
> > >
> > > From the results of the above two correlation metrics, we can conclude that our modality-shared features are indeed modality-shared, as they have a strong positive correlation across modalities. And our modality-specific features are indeed modality-specific, as they have a very weak correlation across modalities.
> > >
> > > If you have any other questions or concerns, we would really appreciate the opportunity to discuss them with you further. Thank you again!

---

> > > > ### Comment · Reviewer_8EtQ · 2023-08-15
> > > >
> > > > Thanks for the authors' reply.
> > > >
> > > > Authors use numbers to validate that "modality-shared" features are close across domains while far away from "modality-specific" features. However, this can not lead to those "modality-shared" features representing shapes, actions, or gestures.
> > > >
> > > > What I intend to state is that it's not guaranteed that "modality-shared feature" in this paper is truly meaningful and sharable. For example, stylegan-like works use example [content A] + [style B] = [image A with style B] to validate [content] feature has extract content information.

---

> > > > > ### Author Response · Authors · 2023-08-16
> > > > >
> > > > > Thanks for proposing your further insightful questions.
> > > > >
> > > > > Following your suggestions, we further analyzed the meaningfulness and the ability to share features of modality-shared features by adding another experiment on cross-modal retrieval (similar to Table 13 in CLIP [1]). The ability of cross-modal retrieval is highly related to the shareable of features, as only features that are meaningful, shareable, and highly connective can give a good recall rate.
> > > > > We use modality-shared features of videos to retrieve from the modality-shared features of audios and calculate the recall, and vice versa for audios. We report the R@1, R@5, and R@10 values for Video to Audio Retrieval and Audio to Video Retrieval. For example, R@5 for Video to Audio Retrieval means we retrieve 5 candidates from audio, if at least one of them has the same label as the query video, we consider the retrieval success and then we calculate the average success rate for all query videos. We also do the same thing on modality-specific features: using modality-specific features of videos to retrieve from the modality-specific features of audios, and vice versa for audios. As shown below, we have a very high recall rate when we use modality-shared features, while the recall rate is biased towards random when we use modality-specific features. This further indicates that the modality-shared features are truly shareable and meaningful and there are very strong relations and connections across modalities. The modality-specific features are instead with more information that is private to each single modality.
> > > > >
> > > > > **Video to Audio Retrieval:**
> > > > > |  | R@1 | R@5 | R@10 |
> > > > > | :--- | :--- | :--- | :---
> > > > > | Modality-specific Features | 11.80 | 45.37 | 49.78 |
> > > > > | Modality-shared Features | $\mathbf{76.58}$ | $\mathbf{90.13}$ | $\mathbf{92.83}$ |
> > > > >
> > > > > **Audio to Video Retrieval:**
> > > > > |  | R@1 | R@5 | R@10 |
> > > > > | :--- | :--- | :--- | :--- |
> > > > > | Modality-specific Features | 10.24 | 33.01 | 47.10 |
> > > > > | Modality-shared Features |  $\mathbf{67.59}$ | $\mathbf{89.60}$ | $\mathbf{93.99}$ |
> > > > >
> > > > > To further verify the meaningfulness of modality-shared features, we also train a classifier only on the concatenation of modality-shared features of video and audio, and discard the modality-specific features. As shown below, without modality-specific features, the performances drop slightly compared to the whole SimMMDG framework, but still competitive. This indicates that modality-shared features truly have some meaningful information that can be used for prediction tasks.
> > > > >
> > > > > | Method  | D2,D3→D1 | D1,D3→D2 | D1,D2→D3 | mean |
> > > > > |---------|----------|----------|----------|------|
> > > > > | SimMMDG (modality-shared features only) |54.71|64.67|62.83|60.74|
> > > > > | SimMMDG |57.93|65.47|66.32|63.24|
> > > > >
> > > > > In this paper, we assume modality-shared features reflect all shared/mutual information between different modalities, rather than specific attributes like identity, pose, or background as in [2]. This is reasonable, as different modalities describing the same action/class/object should have some mutual information, as motivated in our introduction part and Figure 1 (a) in the main paper. We only disentangle features at the modality level, i.e. modality-shared and modality-specific features, to keep the whole framework simple and extendable. We don't further disentangle modality-shared or modality-specific features into different domain-specific attributes, like identity, pose, and background, as in [2].
> > > > > Furthermore, it is important to note that our work primarily focuses on the realm of classification rather than generation. These are the reasons why it is not possible to validate the features in a similar way to StyleGAN. However, we effectively validate that modality-shared features are truly meaningful and shareable and modality-specific features are specific pieces of information that are unique to each modality, by the carefully designed experiments above, as well as the Pearson correlation coefficient and Cosine similarity calculated in the last comment.
> > > > > In future work, it would be also interesting to combine [2] with our proposed framework to learn disentangled features at both domain and modality levels.
> > > > >
> > > > > Thanks for your insightful questions! We will put all added experiments in the main paper for better clarity. If you have any other questions or concerns, we would really appreciate the opportunity to discuss
> > > > > them with you further. Thank you again!
> > > > >
> > > > > [1] Radford, Alec, et al. "Learning transferable visual models from natural language supervision." ICML, 2021.
> > > > >
> > > > > [2] Ge, Yunhao, et al. "Zero-shot synthesis with group-supervised learning." arXiv preprint arXiv:2009.06586 (2020).

---

> > > > > > ### Comment · Reviewer_8EtQ · 2023-08-17
> > > > > >
> > > > > > I appreciate the authors' effort in response.
> > > > > >
> > > > > > Modality-shared feature shows an absolute advantage against the modality-specific feature on cross-modal retrieval tasks. I believe these experiments can prove that the modality-shared features are truly shared across modalities cause it benefits the cross-modal tasks. And this is more convincing than the feature distance of different feature groups (shared-shared, shared-specific, specific-specific feature). I highly recommend including those cross-modal experiments in the main paper.
> > > > > >
> > > > > > I believe the authors have addressed my concerns and I will raise my score to positive.

---

> > > > > > > ### Author Response · Authors · 2023-08-17
> > > > > > > **Thanks for recognizing our efforts and raising your score to positive!**
> > > > > > >
> > > > > > > We are glad to hear that we have addressed your concerns and that you raised your score to positive! Thanks for spending a significant amount of time on our submission and giving lots of valuable and insightful suggestions, which make our paper even stronger! We will for sure include all added experiments in the final paper for better clarification.

---

### Official Review · Reviewer_c1hR · 2023-07-04

**Soundness:** 3 good
**Presentation:** 3 good
**Contribution:** 2 fair
**Rating:** 6
**Confidence:** 4

**Summary:**

This paper addresses the challenge of domain generalization in a multi-modal setting. The core idea is that aligning features from multiple modalities in a shared embedding space can hinder generalization. To overcome this limitation, the paper proposes a more effective approach: it separates the features within each modality into modality-specific and modality-shared components and then aligns only the modality-specific components. This alignment is achieved through supervised contrastive learning and cross-modal feature translation techniques. To validate the proposed approach, the paper introduces a new dataset called HAC, which consists of video, audio, and optical flow data from three domains: Human, Animal, and Cartoon. Extensive experiments conducted on both the HAC dataset and the EPIC-Kitchens dataset demonstrate the impressive performance of the proposed method when compared to existing approaches.

**Strengths:**

1. Multi-modal domain generalization is an important practical problem and the proposed method and dataset introduced in this paper will help in advancing research in this direction.
2. From a methodological standpoint, the idea of splitting the features into modality-specific and modality-shared and aligning them separately seems new. Interestingly, this goes against the strategy followed by popular multi-modal models such as CLIP.
3. The paper showcases a comprehensive range of experiments to validate the effectiveness of the proposed method. It explores various combinations of modalities (e.g., Audio+Video, Audio+Video+OpticalFlow), different numbers of training domains, and even evaluates performance in scenarios with missing modalities during inference. This thorough evaluation provides a robust understanding of the method's capabilities and limitations.
4. The paper is very well written and easy to follow. The diagram in Figure 2 is especially well done and effectively illustrates the framework in a concise manner.

**Weaknesses:**

1. The paper highlights an apparent contradiction in its argument regarding modality-specific information. On one hand, it acknowledges that certain modalities, such as video and flow, possess distinct and unalignable features, such as lighting and motion information. However, the paper introduces a cross-modal transformation module that seemingly enables seamless transformation between these features. This raises a critical question: Does this transformation not contradict the initial premise that modality-specific information is inherently unalignable?
2. The paper notably overlooks the utilization of domain labels or any explicit feature alignment techniques across domains. This omission raises concerns about the effectiveness of the proposed method in achieving robust domain generalization.

**Questions:**

1. Can the usage of domain labels and explicit domain alignment strategies improve the generalization performance?
2. Can you clarify the contradiction pointed out in Weakness-1?

**Limitations:**

The authors adequately addressed the limitations in the paper.

---

> ### Author Rebuttal · Authors · 2023-08-08
>
> Thanks for your insightful reviews, and we appreciate your valuable suggestions! We address your concerns and questions as follows:
>
> >**Q1**: The paper highlights an apparent contradiction in its argument regarding modality-specific information. On one hand, it acknowledges that certain modalities, such as video and flow, possess distinct and unalignable features, such as lighting and motion information. However, the paper introduces a cross-modal transformation module that seemingly enables seamless transformation between these features. This raises a critical question: Does this transformation not contradict the initial premise that modality-specific information is inherently unalignable?
>
> **A1**: We don’t aim at aligning modality-specific features across modalities. Instead, we focus on learning an approximate mapping from the embedding space of one modality to another.
>
> 1. **Cross-modal translation won't undermine the unique features of different modalities.** We want to learn an MLP projection to translate the embedding $\mathbf{E}^{i}$ of the $i$-th modality to the embedding $\mathbf{E}^{j}$ of the $j$-th modality. We add a translation loss to make the translated embedding $\mathbf{E}^{j}_{t}$ to be close to $\mathbf{E}^{j}$, **without any explicit alignment or constraints on $\mathbf{E}^{i}$ and $\mathbf{E}^{j}$.** We apply the cross-modal translation on the integrated feature of each modality $\mathbf{E}^{i} = [\mathbf{E}^{i}_s; \mathbf{E}^{i}_c]$, which is the concatenation of modality-specific feature $\mathbf{E}_s$ and modality-shared feature $\mathbf{E}_c$. We still enforce a distance loss on $\mathbf{E}_s$ and $\mathbf{E}_c$ at the same time. Therefore, the modality-specific and modality-shared features are still forced to be separated during the training progress. The embedding visualization shown in Figure 4 (d) and (e) in the Supplementary Material also indicates that for both video and audio, their modality-specific and modality-shared features are well disentangled and are not influenced by the cross-modal translation module.
>
> 2. **Cross-modal translation is a type of modalities interaction,** where different modality elements interact to give rise to new information when integrated together for task inference [1]. Several works [2,3] have already demonstrated that modalities interaction can help improve the performance of multi-modal tasks. For example, the proposed approach in [2] learns a coordinated similarity space between image and text to improve image classification. The approach proposed in [3] translates language into video and audio for language sentiment analysis.
>
> 3. **Cross-modal translation can be thought of as means to leveraging the information from one modality to infer as much information as possible for the target modality.** Just like when we hear a dog barking, we will fill in the picture of the puppy in our mind, and when we see a foreign language, we will automatically translate it into our native language. Although there is information loss and we can't recover all the details during this translation progress, we can still infer some useful information for the target modality. As shown in our ablation study (Table 5 in the main paper), adding the cross-modal translation module indeed improves multi-modal DG performance.
>
> 4. More importantly, our **cross-modal translation module can be used for improving missing-modality generalization,** by filling in the features of missing modality with the features inferred/translated from the available modality. The benefit of adopting cross-modal translation is demonstrated in Table 4 of the main paper. Our approach is robust even when two modalities out of three are missing. This indicates that the inferred information from the cross-modal translation module is valuable and useful for downstream tasks.
> ___
>
> >**Q2**: The paper notably overlooks the utilization of domain labels or any explicit feature alignment techniques across domains. This omission raises concerns about the effectiveness of the proposed method in achieving robust domain generalization?
>
> **A2**: The main goal of the paper is to propose a general framework for multi-modal (multi-/single- source) DG, since in many real cases domain labels are not available. But the reviewer raises a valid point; when the domain labels are available, we can use explicit feature alignment techniques across domains to improve the generalization ability.
>
> Following your suggestion, we combined SimMMDG with a Domain-adversarial Neural Network (DANN) [4] to align the features of source domains using domain labels. This setup has resulted in a further improvement compared to our proposed setup (without using the domain labels). As suggested by Reviewer kbvb, we also combined SimMMDG with Gradient Blending [53] and observed a similar improvement. This indicates that our proposed SimMMDG can be seamlessly combined with other training strategies due to its general and simplicity to get even better performances.
>
> | Method  | D2,D3→D1 | D1,D3→D2 | D1,D2→D3 | mean |
> |---------|----------|----------|----------|------|
> | SimMMDG |57.93|65.47|66.32|63.24|
> | SimMMDG+Gradient Blending |59.31|68.40|66.63|**64.78**|
> | SimMMDG+DANN |60.69|66.93|64.58|**64.07**|
> ___
> [1] Paul Pu Liang, et al. Foundations and recent trends in multimodal machine learning: Principles, challenges, and open questions, arXiv preprint arXiv:2209.03430, 2022.
>
> [2] Andrea Frome, et al. Devise: A deep visual-semantic embedding model. In NeurIPS, 2013.
>
> [3] Hai Pham, et al. Found in translation: Learning robust joint representations by cyclic translations between modalities. In AAAI, 2019.
>
> [4] Yaroslav Ganin and Victor Lempitsky. Unsupervised domain adaptation by backpropagation. In ICML, 2015.

---

> > ### Comment · Reviewer_c1hR · 2023-08-18
> >
> > Thank you for the response. The authors have adequately addressed my concerns. The explanation for the domain translation makes it more apparent to me. I would encourage the authors to include this intuitive reasoning in the main paper and explicitly point out that this translation is lossy, as explained above. Also, the experiments with domain labels strengthen the method in my opinion. Thus, I would like to raise my score to weak accept.

---

> > > ### Author Response · Authors · 2023-08-18
> > > **Thanks for recognizing our work and being willing to raise your score!**
> > >
> > > We are glad to hear that we have addressed your concerns and that you will raise your score! Thanks for spending a significant amount of time on our submission and giving lots of valuable and insightful suggestions, which make our paper even stronger! We will include all added experiments and intuitive reasoning for the cross-modal translation module in the final paper for better clarification.

---

> > > ### Author Response · Authors · 2023-08-21
> > > **A kind reminder to update the final rating in the system**
> > >
> > > Dear Reviewer c1hR,
> > >
> > > We sincerely appreciate your insightful comments and willingness to raise your score to weak accept! Your feedback has been immensely beneficial in enhancing the quality of our paper.
> > >
> > > Considering the author-reviewer discussion period is ending soon, we are afraid that the rating cannot be changed anymore after this period. Could you please spend some time updating the rating in the system if you are satisfied with our rebuttal? If you have any other questions or concerns, we would really appreciate the opportunity to discuss them with you further.
> > >
> > > Thank you once again for your time and engagement throughout this process!

---

### Official Review · Reviewer_kbvb · 2023-07-05

**Soundness:** 3 good
**Presentation:** 3 good
**Contribution:** 3 good
**Rating:** 6
**Confidence:** 4

**Summary:**

This paper presents a framework for domain generalization for models trained with multi-modal/ multi-source data: SimMMDG. The approach is motivated by the observation that each modality contains features shared with other modalities as well as features specific to that modality. The paper proposes to split per-modality features such that modality-shared features should be similar while modality-specific features should be complementary. A multi-loss strategy is adopted to implement this idea. The author evaluates the approach's performance on EPIC-Kitchen and a proposed  Human-Animal-Cartoon dataset.

**Strengths:**

- The proposed approach, SimMMDG seems simple yet novel. It relies on a simple intuition that modalities should have modality-specific features, which had been overlooked in previous literature.  The approach seems very general and has the potential to become a popular framework used by any researcher working with multi-modal representation learning. Its simplicity also enables combining this framework with previous approaches.
- On the empirical results, the method seems to provide non-trivial gain over baselines and previous approaches targeting multi-modal supervised classification.
- The paper is clearly written; it feels one can directly implement the proposed idea without too much trouble.

**Weaknesses:**

- Despite how much I appreciate the idea proposed by the paper, some important experiments are missing, which leads me to question the technical soundness of the work. For example, the learning rate of CL (supervised contrastive loss) is set to 3.0; this essentially increased the effective learning rate of the model significantly. I wonder if authors have tuned the baseline learning rate accordingly to ensure that the performance gain is not influenced by the learning rate change. In Table5, it seems all results adds CL; what if we remove CL, can CT effectively learn similarly as CL? In addition, the work is often motivated with the recent use of text + vision, but the evaluation does not include any text+vision task.
- Impact of the work. The proposed framework seems very general, but the author eventually chose to test only on Domain Generalization. This seems to undersell the work, since it seems feasible to implement this idea to perform multi-modal classification tasks in a generic fashion. I wonder if authors have tried more evaluation apart from domain generalization, and if the methods provide improvement there.
- Lack of discussion on HAC. One of the contribution of the paper is the HAC dataset. This is a new evaluation benchmark, yet studies/ analysis/ details on this dataset is missing. I also checked the supplementary of the paper, but found very limited information on this benchmark, such as dataset statistics, how are videos selected, how is the annotation performed, is there any QA adopted. Without details and studies (e.g. comprehensive baseline evaluation) on HAC, it will be hard to count HAC as a sound contribution and trust evaluation on this new benchmark.
- SimMMDG with other baseline methods. A major contribution author claims is on the simplicity. This makes me wonder if the pipeline is actually compatible with many of the baseline adopted, such as Gradient Blending [53] and Non-Local [54]. I wonder if authors have tried their approach with these methods to show if the proposed framework is complementary to existing techniques.

In general, although I really enjoy reading this paper and likes the intuitions, it seems the work needs more polish and empirical evaluation to prove its soundness.

**Questions:**

See weaknesses

**Limitations:**

Author should discuss the potential impact regarding their newly proposed dataset HAC.

---

> ### Author Rebuttal · Authors · 2023-08-08
>
> Thanks for your insightful reviews, and we appreciate your valuable suggestions! We address your concerns and questions as follows:
>
> >**Q1**: The learning rate of CL is set to 3.0; this essentially increased the effective learning rate of the model significantly. I wonder if authors have tuned the baseline learning rate accordingly to ensure that the performance gain is not influenced by the learning rate change.
>
> **A1**: We apologize for the confusion. In fact, 3.0 is not the learning rate (lr) of CL, but the weight to balance different loss terms, as indicated in formula (5) in the main paper. For SimMMDG, we choose a lr of 0.0001 for all experiments by grid search. For each baseline method, we also tune the lr using grid search and select the one with the best performance.
> ___
>
> >**Q2**: In Table5, it seems all results adds CL; what if we remove CL, can CT effectively learn similarly as CL?
>
> **A2**: We additionally evaluated the contributions of the supervised contrastive loss (CL) and report the results below.
> |Method|D2,D3→D1|D1,D3→D2|D1,D2→D3|mean|
> |---------|----------|----------|----------|------|
> |SimMMDG w/o CL|54.25|63.47|63.04|60.25|
> |SimMMDG|**57.93**|**65.47**|**66.32**|**63.24**|
>
> Although SimMMDG without CL is already better than most baselines, it still has a performance gap compared with the whole framework, which means CT is helpful but cannot replace CL. The effect of CL is similar to explicit feature alignment. It aligns the modality-shared features of different modalities from different source domains with the same label to be as close as possible in the embedding space while pushing away features with different labels, to make the embedding space more distinctive.
> ___
>
> >**Q3**: In addition, the work is often motivated with the recent use of text + vision, but the evaluation does not include any text+vision task.
>
> **A3**: Text is not a usual modality in DG setup and there is no public benchmark on text-based DG, not to mention multi-modal DG. Therefore, we don't include any text+vision DG task. However, as you suggest, we indeed added several multi-modal classification tasks on text+video+audio below.
> ___
>
> >**Q4**: I wonder if authors have tried more evaluation apart from domain generalization, and if the methods provide improvement there.
>
> **A4**: Thanks for your suggestion! Following your suggestion, we evaluated our framework in a more general multi-modal classification setup without DG. We first evaluated the proposed framework on the EPIC-Kitchens dataset without including the DG setup. For that, we mixed the data from three domains together and split the training, validation, and testing data accordingly, such that there are no apparent domain shifts between training and testing data. The results indicate that our proposed SimMMDG has obvious advantages in the general multi-modal classification setup.
>
> |  |EPIC-Kitchens|
> |---------|----------|
> |DeepAll|64.89|
> |TSN [1]|66.33|
> |TBN [28]|73.18|
> |Gradient Blending [53]|73.97|
> |SimMMDG|**76.42**|
>
> We further evaluate our framework on two multi-modal datasets, namely MUSTARD [2] and UR-FUNNY [3] in MultiBench [4]. These two datasets are for human sentiment analysis and involve language, video, and audio modalities. We use the code base in MultiBench for experiments and implement SimMMDG on that. MultiBench treats human sentiment analysis as a regression task, while our framework is designed for classification. Therefore, we modified the code base to fit the classification task. For all baselines, we chose the same backbone and only change the fusion paradigms. Our SimMMDG shows obvious advantages and outperforms the previous SOTA methods with an average gain of 7.3\% and 1.7\%. This indicates that our SimMMDG can also be used as a general framework for multi-modal classification and is compatible with different combinations of modalities (video+audio+flow, language+video+audio). The results further strengthen the contributions of our proposed approach.
>
> |   | MUSTARD | UR-FUNNY |
> |---------|----------|----------|
> |Late Fusion|61.6|63.6|
> |Low-rank Tensor Fusion [5]|65.2|63.9|
> |MULT [6]|60.9|63.2|
> |SimMMDG|**72.5**|**65.6**|
>
> ___
>
> >**Q5**: Lack of discussion on HAC.
>
> **A5**: Due to space limits, we put more details on our dataset in the **global response** at the top of this page.
> ___
>
> >**Q6**: SimMMDG with other baseline methods.
>
> **A6**: Following your suggestion, we combined SimMMDG with Gradient Blending and the results are further improved. As suggested by Reviewer c1hR, we also combine SimMMDG with a Domain-adversarial Neural Network (DANN) [7] to align the features of source domains using domain labels and observe a similar improvement. This indicates that our SimMMDG can be seamlessly combined with other training strategies due to its general and simplicity to get even better performance improvements.
> | Method  | D2,D3→D1 | D1,D3→D2 | D1,D2→D3 | mean |
> |---------|----------|----------|----------|------|
> | SimMMDG |57.93|65.47|66.32|63.24|
> | SimMMDG+Gradient Blending |59.31|68.40|66.63|**64.78**|
> | SimMMDG+DANN |60.69|66.93|64.58|**64.07**|
> ___
> [1] Limin Wang, et al. Temporal segment networks: Towards good practices for deep action recognition. In ECCV, 2016.
>
> [2] Santiago Castro, et al. Towards multimodal sarcasm detection. In ACL, 2019.
>
> [3] Md Kamrul Hasan, et al. Ur-funny: A multimodal language dataset for understanding humor. In EMNLP-IJCNLP, 2019.
>
> [4] Paul Pu Liang, et al. MultiBench: Multiscale Benchmarks for Multimodal Representation Learning. In NeurIPS, 2021.
>
> [5] Zhun Liu, et al. Efficient low-rank multimodal fusion with modality-specific factors. In ACL, 2018.
>
> [6] Yao-Hung Hubert Tsai, et al. Multimodal transformer for unaligned multimodal language sequences. In ACL, 2019.
>
> [7] Yaroslav Ganin and Victor Lempitsky. Unsupervised domain adaptation by backpropagation. In ICML, 2015.

---

> > ### Comment · Reviewer_kbvb · 2023-08-11
> > **Thanks for your response**
> >
> > After reading authors' rebuttal and other reviews, I think the main concerns are addressed very well with the new experiments provided. I think authors should consider including them in the main draft of the paper. These results improve the soundness of the proposed approach.
> >
> > I am inclined to raise my rating for acceptance. Would like to hear the post-rebuttal feedback from other reviewers.
> >
> > Minor: For Q1, I meant setting weight 3.0 means the learning rate is changed by 3x. But since you did parameter search already, this is not a concern anymore.

---

> > > ### Author Response · Authors · 2023-08-12
> > > **Thanks for being inclined to raise the rating for acceptance**
> > >
> > > We are glad to hear that we have addressed most of your concerns and that you are inclined to raise the rating for acceptance! Thanks for spending a significant amount of time on our submission and giving lots of valuable suggestions, which make our paper even stronger! We will for sure include all added experiments and the details on our HAC dataset in the final paper.

---

> > > ### Author Response · Authors · 2023-08-21
> > > **A kind reminder to update the final rating in the system**
> > >
> > > Dear Reviewer kbvb,
> > >
> > > We sincerely appreciate your insightful comments and willingness to raise your rating for acceptance! Your feedback has been immensely beneficial in enhancing the quality of our paper.
> > >
> > > Considering the author-reviewer discussion period is ending soon, we are afraid that the rating cannot be changed anymore after this period. Could you please spend some time updating the rating in the system if you are satisfied with our rebuttal? If you have any other questions or concerns, we would really appreciate the opportunity to discuss them with you further.
> > >
> > > Thank you once again for your time and engagement throughout this process!

---

### Official Review · Reviewer_Zavq · 2023-07-06

**Soundness:** 3 good
**Presentation:** 3 good
**Contribution:** 3 good
**Rating:** 5
**Confidence:** 2

**Summary:**

This paper proposes a method for multi-model domain generalization. Rather than mapping features of different modality into the same embedding space, it splits features within each modality into modality-specific feature and modality-shared feature. Then it applies different learning strategies to different split features. Besides, a new dataset called Human-Animal-Cartoon is proposed in this paper. Experiments a re conducted on two datasets and shows good performance.

**Strengths:**

The paper proposes a novel method for multi-model domain generalization and a new dataset for this task. The motivation and method are clear to me. The results are convincing to me.

**Weaknesses:**

I am confused to the section 3.2.2 Cross-model Translation. The design is translating the i-th modality feature to j-th modality feature. However, like the paper states that, different modality features contains share-features and unique features. Does the translation will undermine the unique features in different modalities?

**Questions:**

See the weaknesses.

**Limitations:**

The cross-modal translation does not make sense to me. Will rise my rating if rebuttal can address my concern.

---

> ### Author Rebuttal · Authors · 2023-08-08
>
> Thanks for your insightful reviews, and we appreciate your valuable suggestions! We address your concerns and questions as follows:
>
> > **Q1:** I am confused to the section 3.2.2 cross-modal Translation. The design is translating the i-th modality feature to j-th modality feature. However, like the paper states that, different modality features contains share-features and unique features. Does the translation will undermine the unique features in different modalities?
>
> **A1:**
>
> 1. **Cross-modal translation won't undermine the unique features of different modalities.** We want to learn an MLP projection to translate the embedding $\mathbf{E}^{i}$ of the $i$-th modality to the embedding $\mathbf{E}^{j}$ of the $j$-th modality. We add a translation loss to make the translated embedding $\mathbf{E}^{j}_{t}$ to be close to $\mathbf{E}^{j}$, without any explicit alignment or constraints on $\mathbf{E}^{i}$ and $\mathbf{E}^{j}$. We apply the cross-modal translation on the integrated feature of each modality $\mathbf{E}^{i} = [\mathbf{E}^{i}_s; \mathbf{E}^{i}_c]$, which is the concatenation of modality-specific feature $\mathbf{E}_s$ and modality-shared feature $\mathbf{E}_c$. We still enforce a distance loss on $\mathbf{E}_s$ and $\mathbf{E}_c$ at the same time. Therefore, the modality-specific and modality-shared features are still forced to be separated during the training progress. The embedding visualization shown in Figure 4 (d) and (e) in the Supplementary Material also indicates that for both video and audio, their modality-specific and modality-shared features are well disentangled and are not influenced by the cross-modal translation module.
>
> 2. **Cross-modal translation is a type of modalities interaction,** where different modality elements interact to give rise to new information when integrated together for task inference [1]. Several works [2,3] have already demonstrated that modalities interaction can help improve the performance of multi-modal tasks. For example, the proposed approach in [2] learns a coordinated similarity space between image and text to improve image classification. The approach proposed in [3] translates language into video and audio for language sentiment analysis.
>
> 3. **Cross-modal translation can be thought of as means to leveraging the information from one modality to infer as much information as possible for the target modality.** Just like when we hear a dog barking, we will fill in the picture of the puppy in our mind, and when we see a foreign language, we will automatically translate it into our native language. Although there is information loss and we can't recover all the details during this translation progress, we can still infer some useful information for the target modality. As shown in our ablation study (Table 5 in the main paper), adding the cross-modal translation module indeed improves multi-modal DG performance.
>
> 4. More importantly, our **cross-modal translation module can be used for improving missing-modality generalization,** by filling in the features of missing modality with the features inferred/translated from the available modality. The benefit of adopting cross-modal translation is demonstrated in Table 4 of the main paper. Our approach is robust even when two modalities out of three are missing. This indicates that the inferred information from the cross-modal translation module is valuable and useful for downstream tasks.
>
> [1] Paul Pu Liang, et al. Foundations and recent trends in multimodal machine learning: Principles, challenges, and open questions, arXiv preprint arXiv:2209.03430, 2022.
>
> [2] Andrea Frome, et al. Devise: A deep visual-semantic embedding model. In NeurIPS, 2013.
>
> [3] Hai Pham, et al. Found in translation: Learning robust joint representations by cyclic translations between modalities. In AAAI, 2019.

---

> > ### Comment · Reviewer_Zavq · 2023-08-18
> > **Thanks for the response**
> >
> > Thank authors for the detailed clarification. It make sense to me. After reading all other reviewer's opinion, I will change to my rating to accept.

---

> > > ### Author Response · Authors · 2023-08-18
> > > **Thanks for recognizing our work and being willing to change your rating to accept!**
> > >
> > > We are glad to hear that we have addressed your concerns and that you will change your rating to accept! Thanks for spending a significant amount of time on our submission and giving lots of valuable and insightful suggestions, which make our paper even stronger! We will also include all added experiments and points in the final paper for better clarification.

---

> > > ### Author Response · Authors · 2023-08-21
> > > **A kind reminder to update the final rating in the system**
> > >
> > > Dear Reviewer Zavq,
> > >
> > > We sincerely appreciate your insightful comments and willingness to change your rating to accept! Your feedback has been immensely beneficial in enhancing the quality of our paper.
> > >
> > > Considering the author-reviewer discussion period is ending soon, we are afraid that the rating cannot be changed anymore after this period. Could you please spend some time updating the rating in the system if you are satisfied with our rebuttal? If you have any other questions or concerns, we would really appreciate the opportunity to discuss them with you further.
> > >
> > > Thank you once again for your time and engagement throughout this process!

---

### Official Review · Reviewer_93nE · 2023-07-06

**Soundness:** 3 good
**Presentation:** 3 good
**Contribution:** 3 good
**Rating:** 6
**Confidence:** 4

**Summary:**

The paper tackles the problem of multimodal domain generalization in which the goal is to learn a model from multi-modal multi/single-source domain data that can generalize to unseen multimodal distributions. The main motivation is that mapping data from multiple modalities to a shared embedding space only harnesses the common features among different modalities while overlooks the modality-specific information. The paper proposes three main technical contributions to tackle the aforementioned problem and address the multimodal domain generalization problem. At first, each modality features are split into modality-shared and modality-specific components and then a supervised contrastive loss is used for alignment. Next, a distance loss is leveraged to repel the modality-specific features from each modality. Also, a cross-modal translation module is proposed to regularize the learned features and facilitate the missing modality issue. Finally, the paper also introduces a new multimodal DG dataset, namely Human-Animal-Cartoon (HAC). Results on EPIC-kitchens and HAC datasets claim improved DG performance under both multi-source and single-source settings.

**Strengths:**

1) The problem of multimodal domain generalization is relatively new and mostly underxplored and carries practical relevance because in many real-world scenarios the data is often observed via multiple modalities.

2) The cross-modal translation module is capable of regularizing the learned features and handle missing modalities issue.

3) The paper proposes a new Human-Animal-Cartoon (HAC) dataset, which consists of different action classes and three different domains, to facilitate research in multimodal DG.

4) Results on two different datasets and under both multi-source and single-source DG setting show that the proposed framework provides notable gains over the baselines and existing methods.

**Weaknesses:**

1) The paper seem to be missing the domain generalization aspect, like which pertinent DG problem in the context of multimodal data the paper is trying to solve, because all the technical contributions mostly aim at either bringing the modality-shared component or repel the modality-specific component which seems effective compared to the only relevant baseline [44].

2) The two important technical contributions i.e. learning shared embedding space and pushing modality-specific feature is not very new in the context of DG and even broader scope [A],[B]. This is also because the supervised contrastive loss [29] as well as the distance based loss are the existing components taken from literature.

3) The unimodal DG comparison is made with rather old approaches RSC [23] and MixUp [55] which weakens the claim of significant performance gain. There are several new unimodal DG approaches which have shown state-of-the-art performances such as [C], [D] and [E].

4) Fig. 2 that displays the overall architecture of the proposed method is not very helpful towards grasping the mid-level idea. For instance, one thing is to improve the patterns and colors. Also, the some symbols from the text in methodology should be used in the diagram to better connect the two.

5) Details on how the HAC datsets is being developed is missing. For instance, how many volunteers were involved? what were the quality assurance mechanisms in place? I was expecting these details at least in the supplementary material, but the supp material only provides some high-level statistics of the dataset such as number of videos or to a little bit class-imbalance issue.

[A] Bui, M.H., Tran, T., Tran, A. and Phung, D., 2021. Exploiting domain-specific features to enhance domain generalization. Advances in Neural Information Processing Systems, 34, pp.21189-21201.

[B] Chattopadhyay, P., Balaji, Y. and Hoffman, J., 2020. Learning to balance specificity and invariance for in and out of domain generalization. In Computer Vision–ECCV 2020: 16th European Conference, Glasgow, UK, August 23–28, 2020, Proceedings, Part IX 16 (pp. 301-318). Springer International Publishing.

[C] Cha, J., Chun, S., Lee, K., Cho, H.C., Park, S., Lee, Y. and Park, S., 2021. Swad: Domain generalization by seeking flat minima. Advances in Neural Information Processing Systems, 34, pp.22405-22418.

[D] Rame, A., Kirchmeyer, M., Rahier, T., Rakotomamonjy, A., Gallinari, P. and Cord, M., 2022. Diverse weight averaging for out-of-distribution generalization. Advances in Neural Information Processing Systems, 35, pp.10821-10836.

[E] Rame, A., Dancette, C. and Cord, M., 2022, June. Fishr: Invariant gradient variances for out-of-distribution generalization. In International Conference on Machine Learning (pp. 18347-18377). PMLR.

**Questions:**

The paper tackles a challenging and a relatively new problem of how to learn a cross-domain generalizable model under multi-modal settings. Overall, the paper presents an effective framework comprised of three components and alongside a new multimodal dataset, however, there are some important questions/concerns as listed in weaknesses (1-5) due to which my initial rating for the paper is ‘weak accept’.

Also:

It would be interesting to know the performance of the method under:

a) different types of domain shifts, such as texture, background etc.

b) increasing domain shifts.

Are the results reported as the average of different trials and if yes, what are the standard deviations?







**Limitations:**

The paper mentions one limitation regarding the increasing complexity of cross-modal translation module with increasing number of modalities.

---

> ### Author Rebuttal · Authors · 2023-08-08
>
> Thanks for your insightful reviews, and we appreciate your valuable suggestions! We address your concerns and questions as follows:
> >**Q1**: The paper seem to be missing the domain generalization aspect, like which pertinent DG problem in the context of multimodal data the paper is trying to solve, because all the technical contributions mostly aim at either bringing the modality-shared component or repel the modality-specific component which seems effective compared to the only relevant baseline [44].
>
> **A1**: This paper aims to solve the general multi-modal DG problem, an extension of traditional DG with multiple modalities as input. In this setup, we train the model on source domains with at least two modalities as input data, and test on target domains that have the same modalities but have large distribution shifts compared to the source domains.
>
> Due to the simplicity of our framework and the proposed cross-modal translation module, our method can also be extended to multi-modal single-source DG and missing-modality DG directly without any modification.
>
> Multi-modal DG is an under-explored problem. [44] is the only known work trying to solve it and is, therefore, the only relevant baseline. However, we also selected several strong multi-modal learning baselines, such as Gradient Blending [53] and MM-SADA [40], in order to verify the effectiveness of the proposed framework.
> ___
> >**Q2**: The two important technical contributions i.e. learning shared embedding space and pushing modality-specific feature is not very new in the context of DG and even broader scope [A],[B]. This is also because the supervised contrastive loss [29] as well as the distance based loss are the existing components taken from literature.
>
> **A2**: [A,B] learn invariant and specific features at the domain level, while our proposed approach learns at the modality level. This is a key difference compared to previous schemes and, thus, a major contribution of this work. In future work, it would be also interesting to combine [A,B] with our proposed framework to learn disentangled features at both domain and modality levels.
>
> Traditional multi-modal learning frameworks use contrastive loss [29] on integrated features of different modalities, while our proposed approach only uses contrastive loss on modality-shared features and adds a distance loss on modality-specific features in order to learn complementary information from different modalities.
>
> In addition to these contributions, we address the missing-modality generalization problem with our proposed cross-modal translation module, which is robust even in scenarios where multiple modalities are missing. Moreover, we also provide theoretical insights in support of the efficacy of our proposed approach.
> ___
>
> >**Q3**: The unimodal DG comparison is made with rather old approaches RSC [23] and MixUp [55] which weakens the claim of significant performance gain. There are several new unimodal DG approaches which have shown state-of-the-art performances such as [C], [D] and [E].
>
> **A3**: In response to the reviewer's comment, we add experiments with one of the suggested SOTA unimodal DG baselines: Fishr [E] (ICML, 2022), as shown in the table below. Fishr outperforms RSC and Mixup in most cases. However, it still yields a large classification accuracy gap with an average of 7.74\% compared to our multi-modal DG method. These additional results further emphasize the contribution of our proposed multi-modal DG framework.
>
> | Method  | D2,D3→D1 | D1,D3→D2 | D1,D2→D3 | mean |
> |---------|----------|----------|----------|------|
> | RSC (V) |50.11    |      62.53    |       58.73   |  57.12    |
> | RSC (A) |36.55    |    43.73      |      48.15    |    42.81  |
> | RSC (F) |55.17   |        63.33  |     59.65     |  59.38    |
> | Mixup  (V) |49.20     |     59.73     |    59.96      |    56.30  |
> | Mixup  (A) |35.17      |        40.80  |     45.07     |   40.35   |
> | Mixup  (F) |56.32    |    65.60      |      54.62    |     58.85 |
> | Fishr (V) |53.79   |     63.47     |    61.09      |  59.45    |
> | Fishr (A) |37.47   |   44.80       |   47.43       |    43.23  |
> | Fishr (F) |54.25    |   63.87       |      59.14    |   59.09   |
> | SimMMDG(V+A+F) |      **63.68**    |     **70.13**     |    **67.76**      |  **67.19**    |
>
> ___
>
> >**Q4**: Fig. 2 that displays the overall architecture of the proposed method is not very helpful towards grasping the mid-level idea. For instance, one thing is to improve the patterns and colors. Also, the some symbols from the text in methodology should be used in the diagram to better connect the two.
>
> **A4**: Thanks for your suggestion! We will indeed change the patterns and colors and add the text symbols in the final version of the paper for better illustration.
> ___
>
> >**Q5**: Details on how the HAC datsets is being developed is missing.
>
> **A5**: Due to space limits, we put more details on our HAC dataset in the **global response** at the top of this page.
> ___
>
> >**Q6**: It would be interesting to know the performance of the method under: a) different types of domain shifts, such as texture, background etc. b) increasing domain shifts. Are the results reported as the average of different trials and if yes, what are the standard deviations?
>
> **A6**: In fact, the EPIC-kitchens and HAC datasets suffer different types of domain shifts. For EPIC-kitchens, the domain shifts come from different backgrounds, e.g. three kitchens from distinct countries with different layouts. The shifts of the HAC dataset mostly come from different textures, e.g. cartoons vs real-world videos. Our SimMMDG shows strong results under different types of domain shifts. Indeed, the results are reported as the average of three different trials (Table 7 in Supplementary Material) with the standard deviations lying within reasonable values, which indicates that our framework has statistical improvements compared to other baselines.

---

> ### Comment · Reviewer_93nE · 2023-08-11
>
> I thank authors for providing a thorough response to my comments in the rebuttal and I have gone over other reviews. Most of the rebuttal responses are satisfactory, and I'm leaning towards accepting the paper, however, I would like more clarification on the following:
>
> Q1: I think this response is bit weak. Because what do you mean by solving the general multi-modal DG problem? It would be good to know the specific challenge(s) that arise when extending traditional DG to multimodal case and this paper is trying to solve?
>
> Q5: Thanks for providing these details. Is it possible to provide any statistics on the dataset development process, for instance, discarded unqualified data, noisy/wrong classes etc. and accuracy of the dataset?

---

> > ### Author Response · Authors · 2023-08-12
> >
> > We are glad to hear that we have addressed most of your concerns and that you are leaning toward accepting the paper! Thanks for spending a significant amount of time on our submission and giving lots of valuable suggestions, which make our paper even stronger! We will include all added experiments and the details on our HAC dataset in the final paper. We address your further concerns and questions as follows:
> > ___
> > **A1**:  There are two main challenges that arise when extending traditional DG to multimodal cases.
> >
> > 1. The first challenge is **how to better exploit complementary information from different modalities.** Different modalities consist of both shared information that is consistent across modalities and unique information that is specific to each modality. It is essential to extract both types of information in order to make better predictions, as motivated in our introduction part and Figure 1 (a) in the main paper. Traditional multi-modal contrastive learning frameworks [29, 45] project the features of different modalities into a common embedding space, which will only preserve modality-shared information and overlook modality-specific information. In order to address this challenge, we propose to split the feature embedding of each modality into modality-specific and modality-shared components and use supervised contrastive learning and distance loss to extract meaningful separated features.
> >
> > 2. The second challenge is **how to effectively facilitate modality interactions.** Modality interactions referred to as different modality elements interact to give rise to new information when integrated together for task inference [1]. Several works [2,3] have already demonstrated that modalities interaction can help improve the performance of multi-modal tasks. For example, the proposed approach in [2] learns a coordinated similarity space between image and text to improve image classification. The approach proposed in [3] translates language into video and audio for language sentiment analysis. In this work, we propose to use a cross-modal translation module for modality interaction to regularize the learned features. We also address the missing-modality generalization problem with our proposed cross-modal translation module, which is robust even in scenarios where multiple modalities are missing.
> >
> > Solving the general multi-modal DG problem means we propose a simple and universal framework, that can be used for multi-modal multi-source DG, multi-modal single-source DG, and missing-modality DG problems. Our framework can also be easily combined with other DG strategies to achieve further performance improvement (SimMMDG+Gradient Blending and SimMMDG+DANN as shown in global response).
> > ___
> > **A5:**  The discarded unqualified data referred to those videos with no audio or with very large background noise, videos with very low resolution and quality, or duplicate videos. Since we introduced the basic requirements for the data to each volunteer beforehand and split the action classes and cartoon series for each person, there is very little such data and we discarded about 10 such unqualified samples.
> >
> > The noisy/wrong classes referred to those videos with ambiguous classes, such as "a person is watching TV and at the same time eating/drinking something", or "a person is running while opening the door". We discarded about 30 such unqualified samples.
> >
> > After these quality assurance mechanisms, we make sure that all data samples are unique and of good quality in our dataset.
> >
> > ___
> > If you have any other questions or concerns, we would really appreciate the opportunity to discuss them with you further. Thank you again!
> > ___
> >
> > [1] Paul Pu Liang, et al. Foundations and recent trends in multimodal machine learning: Principles, challenges, and open questions, arXiv preprint arXiv:2209.03430, 2022.
> >
> > [2] Andrea Frome, et al. Devise: A deep visual-semantic embedding model. In NeurIPS, 2013.
> >
> > [3] Hai Pham, et al. Found in translation: Learning robust joint representations by cyclic translations between modalities. In AAAI, 2019.

---

> > > ### Comment · Reviewer_93nE · 2023-08-18
> > >
> > > I thank authors for providing further clarifications on A1 and A5. I believe authors have satisfactorily addressed my concerns and I would like to keep my pre-rebuttal rating.
> > > I would strongly encourage authors to include the points from rebuttal, including the Q3 experimental results in the main paper.

---

> > > > ### Author Response · Authors · 2023-08-18
> > > > **Thanks for recognizing our efforts and keeping your positive rating!**
> > > >
> > > > We are glad to hear that we have addressed your concerns and that you keep your positive rating! Thanks for spending a significant amount of time on our submission and giving lots of valuable and insightful suggestions, which make our paper even stronger! We will for sure include all added experiments and points in the final paper for better clarification.

---

### Author Rebuttal · Authors · 2023-08-08

>**Q1**: More details on HAC dataset.

**A1**: The detailed dataset statistics such as the name of each action, the number of each action segment, and training-testing splits are in the supplementary material. Here, we provide more details on how we collect the dataset.

Our dataset was collected by 5 volunteers. For the human domain, we collect the data by selecting actions from an existing Kinetics-600 dataset. We select approximately the same number of video clips for each action to ensure class balance. For the animal domain,  we collect data from YouTube by searching keywords like ‘animal sleeping’, 'animal eating', 'animal running', etc. To increase the diversity of the dataset, we also specify the animal type in the keywords, such as ‘cat sleeping’, 'dog eating', 'horse running', etc.  Each participant was asked to collect certain actions, such as participant A for ‘sleeping’ and ‘watching tv’, participant B for ‘eating’ and ‘drinking’, etc. For the cartoon domain, we collect data from popular cartoons like 'SpongeBob SquarePants', 'The Simpsons', 'Garfield and Friends', etc. Each participant was asked to collect from one or two cartoon series to avoid duplication.

For each action, we annotated the start and end times in the video and then cut out video clips. The length of each video clip varies from 1s to 10s. Finally, we gathered the data from all volunteers and a separate person manually discarded unqualified data like duplicate videos, videos without audio data, noisy/wrong classes, etc.

The main purpose of this dataset is to be used for multi-modal domain generalization research. Of course, our dataset can also be used for other applications like multi-modal learning and multi-modal domain adaptation. For comprehensive baseline evaluation on our new dataset, we also provided selected multi-modal DG baseline evaluations in the main paper (DeepAll, MM-SADA, RNA-Net, and SimMMDG). Here, we provide additional evaluations on two baselines including TSN[1] and TBN[28].

||A,C→H|H,C→A|H,A→C|mean|
|---------|----------|----------|----------|------|
|TSN[1]|63.23|71.96|41.82|59.00|
|TBN[28]|70.65|75.71|47.47|64.61|
___
>**Q2: More strengths of SimMMDG framework.**

**A2**: We have a deeper study on our SimMMDG framework based on the comments by Reviewer kbvb and c1hR and report our interesting findings here.

1. **SimMMDG is a general framework for multi-modal classification tasks, not limited to domain generalization setup.** Following the suggestion from Reviewer kbvb, we evaluated our framework in a more general multi-modal classification setup without DG. We first evaluated the proposed framework on the EPIC-Kitchens dataset (video+audio) without including the DG setup. For that, we mixed the data from three domains together and split the training, validation, and testing data accordingly, such that there are no apparent domain shifts between training and testing data. The results indicate that our proposed SimMMDG has obvious advantages in the general multi-modal classification setup.

|  |EPIC-Kitchens|
|---------|----------|
|DeepAll|64.89|
|TSN [1]|66.33|
|TBN [28]|73.18|
|Gradient Blending [53]|73.97|
|SimMMDG|**76.42**|

We further evaluate our framework on two multi-modal datasets, namely MUSTARD [2] and UR-FUNNY [3] in MultiBench [4]. These two datasets are for human sentiment analysis and involve language, video, and audio modalities. We use the code base in MultiBench for experiments and implement SimMMDG on that. MultiBench treats human sentiment analysis as a regression task, while our framework is designed for classification. Therefore, we modified the code base to fit the classification task. For all baselines, we chose the same backbone and only change the fusion paradigms. Our SimMMDG shows obvious advantages and outperforms the previous SOTA methods with an average gain of 7.3\% and 1.7\%. This indicates that our SimMMDG can also be used as a general framework for multi-modal classification and is compatible with different combinations of modalities (video+audio+flow, language+video+audio). The results further strengthen the contributions of our proposed approach.

|   | MUSTARD | UR-FUNNY |
|---------|----------|----------|
|Late Fusion|61.6|63.6|
|Low-rank Tensor Fusion [5]|65.2|63.9|
|MULT [6]|60.9|63.2|
|SimMMDG|**72.5**|**65.6**|

2. **SimMMDG can be seamlessly combined with other training strategies to achieve better performances.** Following the suggestion from Reviewer kbvb, we combined SimMMDG with Gradient Blending and the results are further improved. As suggested by Reviewer c1hR, we also combine SimMMDG with a Domain-adversarial Neural Network (DANN) [7] to align the features of source domains using domain labels and observe a similar improvement. This indicates that our SimMMDG can be seamlessly combined with other training strategies due to its general and simplicity to get even better performance improvements.
| Method  | D2,D3→D1 | D1,D3→D2 | D1,D2→D3 | mean |
|---------|----------|----------|----------|------|
| SimMMDG |57.93|65.47|66.32|63.24|
| SimMMDG+Gradient Blending |59.31|68.40|66.63|**64.78**|
| SimMMDG+DANN |60.69|66.93|64.58|**64.07**|
___
[1] Limin Wang, et al. Temporal segment networks: Towards good practices for deep action recognition. In ECCV, 2016.

[2] Santiago Castro, et al. Towards multimodal sarcasm detection. In ACL, 2019.

[3] Md Kamrul Hasan, et al. Ur-funny: A multimodal language dataset for understanding humor. In EMNLP-IJCNLP, 2019.

[4] Paul Pu Liang, et al. MultiBench: Multiscale Benchmarks for Multimodal Representation Learning. In NeurIPS, 2021.

[5] Zhun Liu, et al. Efficient low-rank multimodal fusion with modality-specific factors. In ACL, 2018.

[6] Yao-Hung Hubert Tsai, et al. Multimodal transformer for unaligned multimodal language sequences. In ACL, 2019.

[7] Yaroslav Ganin and Victor Lempitsky. Unsupervised domain adaptation by backpropagation. In ICML, 2015.

---

### Decision · Program_Chairs · 2023-09-21

**Decision:**

Accept (poster)

**Comment:**

The topic of multi-modal domain generalization is considered of interest, the effectiveness of the simple method is appreciated, and the proposed HAC dataset is valuable. The author-responses to the remaining reviewer concerns have been well received and post-rebuttal all reviewers are supportive of acceptance. The AC agrees and encourages the authors to include all the promised additions and experiments in the camera-ready paper.